# Going Beyond Linear Mode Connectivity:
# The Layerwise Linear Feature Connectivity

**Zhanpeng Zhou[1], Yongyi Yang[2], Xiaojiang Yang[1], Junchi Yan[1],[*] Wei Hu[2][†]**

[1] Dept. of Computer Science and Engineering & MoE Key Lab of AI, Shanghai Jiao Tong University
[2] Dept. of Electrical Engineering & Computer Science, University of Michigan
{zzp1012,yangxiaojiang,yanjunchi}@sjtu.edu.cn
{yongyi,vvh}@umich.edu

## Abstract

Recent work has revealed many intriguing empirical phenomena in neural network training, despite the poorly understood and highly complex loss landscapes and training dynamics. One of these phenomena, Linear Mode Connectivity (LMC), has gained considerable attention due to the intriguing observation that different solutions can be connected by a linear path in the parameter space while maintaining near-constant training and test losses. In this work, we introduce a stronger notion of linear connectivity, *Layerwise Linear Feature Connectivity (LLFC)*, which says that the feature maps of every layer in different trained networks are also linearly connected. We provide comprehensive empirical evidence for LLFC across a wide range of settings, demonstrating that whenever two trained networks satisfy LMC (via either spawning or permutation methods), they also satisfy LLFC in nearly all the layers. Furthermore, we delve deeper into the underlying factors contributing to LLFC, which reveal new insights into the permutation approaches. The study of LLFC transcends and advances our understanding of LMC by adopting a feature-learning perspective. We released our source code at https://github.com/zzp1012/LLFC.

## 1   Introduction

Despite the successes of modern deep neural networks, theoretical understanding of them still lags behind. Efforts to understand the mechanisms behind deep learning have led to significant interest in exploring the loss landscapes and training dynamics. While the loss functions used in deep learning are often regarded as complex black-box functions in high dimensions, it is believed that these functions, particularly the parts encountered in practical training trajectories, contain intricate benign structures that play a role in facilitating the effectiveness of gradient-based training [36, 4, 39]. Just like in many other scientific disciplines, a crucial step toward formulating a comprehensive theory of deep learning lies in meticulous empirical investigations of the learning pipeline, intending to uncover quantitative and reproducible nontrivial phenomena that shed light on the underlying mechanisms.

One intriguing phenomenon discovered in recent work is Mode Connectivity [10, 5, 11]: Different optima found by independent runs of gradient-based optimization are connected by a simple path in the parameter space, on which the loss or accuracy is nearly constant. This is surprising as different optima of a non-convex function can very well reside in different and isolated "valleys" and yet this does not happen for optima found in practice. More recently, an even stronger form of mode

---

[*]Corresponding author. SJTU authors are partly supported by NSFC(62222607, 61972250, U19B2035) and Shanghai Municipal Science and Technology Major Project (2021SHZDZX0102).

[†]Wei Hu acknowledges support from the Google Research Scholar Program.

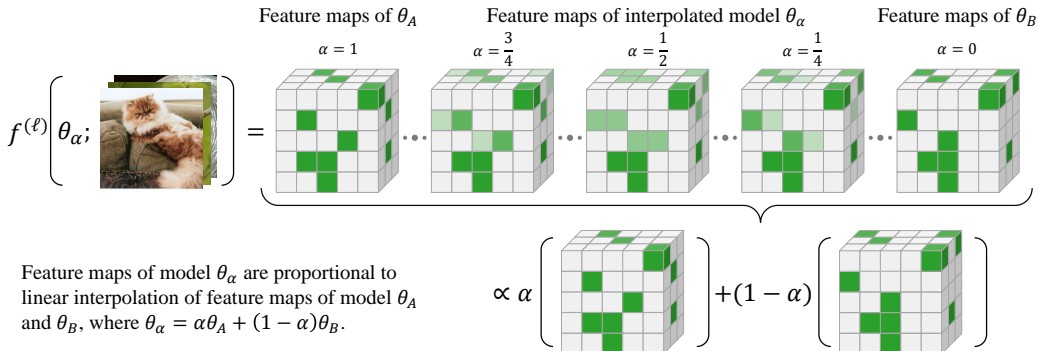

Figure 1: Illustration of Layerwise Linear Feature Connectivity (LLFC). Each tensor comprises three feature maps when provided with three input images.

connectivity called Linear Mode Connectivity (LMC) was discovered [24, 9]. It depicts that different optima can be connected by a *linear* path of constant loss/accuracy. Although LMC typically does not happen for two independently trained networks, it has been consistently observed in the following scenarios:

• **Spawning** [9, 7]: A network is randomly initialized, trained for a small number of epochs (e.g. 5 epochs for both ResNet-20 and VGG-16 on the CIFAR-10 dataset), and then spawned into two copies which continue to be independently trained using different SGD randomnesses (i.e., for mini-batch order and data augmentation).

• **Permutation** [6, 1]: Two networks are independently trained, and the neurons of one model are permuted to match the neurons of the other model while maintaining a functionally equivalent network.

The study of LMC is highly motivated due to its ability to unveil nontrivial structural properties of loss landscapes and training dynamics. Furthermore, LMC has significant relevance to various practical applications, such as pruning and weight-averaging methods.

On the other hand, the success of deep neural networks is related to their ability to learn useful *features*, or *representations*, of the data [27], and recent work has highlighted the importance of analyzing not only the final outputs of a network but also its intermediate features [20]. However, this crucial perspective is absent in the existing literature on LMC and weight averaging. These studies typically focus on interpolating the weights of different models and examining the final loss and accuracy, without delving into the internal layers of the network.

In this work, we take a feature-learning perspective on LMC and pose the question: what happens to the internal features when we linearly interpolate the weights of two trained networks? Our main discovery, referred to as *Layerwise Linear Feature Connectivity (LLFC)*, is that the features in almost all the layers also satisfy a strong form of linear connectivity: the feature map in the weight-interpolated network is approximately the same as the linear interpolation of the feature maps in the two original networks. More precisely, let $\boldsymbol{\theta}_A$ and $\boldsymbol{\theta}_B$ be the weights of two trained networks, and let $f^{(\ell)}(\boldsymbol{\theta})$ be the feature map in the $\ell$-th layer of the network with weights $\boldsymbol{\theta}$. Then we say that $\boldsymbol{\theta}_A$ and $\boldsymbol{\theta}_B$ satisfy LLFC if

$$f^{(\ell)}(\alpha\boldsymbol{\theta}_A + (1-\alpha)\boldsymbol{\theta}_B) \propto \alpha f^{(\ell)}(\boldsymbol{\theta}_A) + (1-\alpha)f^{(\ell)}(\boldsymbol{\theta}_B), \qquad \forall \alpha \in [0,1], \forall \ell. \qquad (1)$$

See Figure 1 for an illustration. While LLFC certainly cannot hold for two arbitrary $\boldsymbol{\theta}_A$ and $\boldsymbol{\theta}_B$, we find that it is satisfied whenever $\boldsymbol{\theta}_A$ and $\boldsymbol{\theta}_B$ satisfy LMC. We confirm this across a wide range of settings, as well as for both spawning and permutation methods that give rise to LMC.

LLFC is a much finer-grained characterization of linearity than LMC. While LMC only concerns loss or accuracy, which is a single scalar value, LLFC (1) establishes a relation for all intermediate feature maps, which are high-dimensional objects. Furthermore, it is not difficult to see that LLFC applied to the output layer implies LMC when the two networks have small errors (see Lemma 1); hence, LLFC can be viewed as a strictly stronger property than LMC. The consistent co-occurrence of LLFC and LMC suggests that studying LLFC may play a crucial role in enhancing our understanding of LMC.

Subsequently, we delve deeper into the underlying factors contributing to LLFC. We identify two critical conditions, *weak additivity for ReLU function* and a *commutativity property* between two trained networks. We prove that these two conditions collectively imply LLFC in ReLU networks, and provide empirical verification of these conditions. Furthermore, our investigation yields novel insights into the permutation approaches: we interpret both the activation matching and weight matching objectives in Git Re-Basin [1] as ways to ensure the satisfaction of commutativity property.

In summary, our work unveils a richer set of phenomena that go significantly beyond the scope of LMC, and our further investigation provides valuable insights into the underlying mechanism behind LMC. Our results demonstrate the value of opening the black box of neural networks and taking a feature-centric viewpoint in studying questions related to loss landscapes and training dynamics.

## 2 Related Work

**(Linear) Mode Connectivity.** Freeman and Bruna [10], Draxler et al. [5], Garipov et al. [11] observed Mode Connectivity, i.e., different optima/modes of the loss function can be connected through a non-linear path with nearly constant loss. Nagarajan and Kolter [24] first observed Linear Mode Connectivity (LMC), i.e., the near-constant-loss connecting path can be linear, on models trained on MNIST starting from the same random initialization. Later, Frankle et al. [9] first formally defined and thoroughly investigate the LMC problem. Frankle et al. [9] observed LMC on harder datasets, for networks that are jointly trained for a short amount of time before going through independent training (we refer to this as the spawning method). Frankle et al. [9] also demonstrated a connection between LMC and the Lottery Ticket Hypothesis [8]. Fort et al. [7] used the spawning method to explore the connection between LMC and the Neural Tangent Kernel dynamics. Lubana et al. [23] studied the mechanisms of DNNs from mode connectivity and verified the mechanistically dissimilar models cannot be linearly connected. Yunis et al. [38] showed that LMC also extends beyond two optima and identified a high-dimensional convex hull of low loss between multiple optima. On the theory side, several papers [10, 21, 32, 26, 25, 16] were able to prove non-linear mode connectivity under various settings, but there has not been a theoretical explanation of LMC to our knowledge.

**Permutation Invariance.** Neural network architectures contain permutation symmetries [13]: one can permute the weights in different layers while not changing the function computed by the network. Ashmore and Gashler [2] utilized the permutation invariance of DNNs to align the topological structure of two neural networks. Tatro et al. [31] used permutation invariance to align the neurons of two neural networks, resulting in improved non-linear mode connectivity. In the context of LMC, Entezari et al. [6], Ainsworth et al. [1] showed that even independently trained networks can be linearly connected when permutation invariance is taken into account. In particular, Ainsworth et al. [1] approached the neuron alignment problem by formulating it as bipartite graph matching and proposed two matching methods: activation matching and weight matching. Notably, Ainsworth et al. [1] achieved the LMC between independently trained ResNet models on the CIFAR-10 dataset using weight matching.

**Model Averaging Methods.** LMC also has direct implications for model averaging methods, which are further related to federated learning and ensemble methods. Wang et al. [33] introduced a novel federated learning algorithm that incorporates unit permutation before model averaging. Singh and Jaggi [30], Liu et al. [22] approached the neuron alignment problem in model averaging by formulating it as an optimal transport and graph matching problem, respectively. Wortsman et al. [35] averaged the weights of multiple fine-tuned models trained with different hyper-parameters and obtained improved performance.

## 3 Background and Preliminaries

**Notation and Setup.** Denote $[k] = \{1, 2, \ldots, k\}$. We consider a classification dataset $\mathcal{D} = \{(\boldsymbol{x}_i, y_i)\}_{i=1}^n$, where $\boldsymbol{x}_i \in \mathbb{R}^{d_0}$ represents the input and $y_i \in [c]$ represents the label of the $i$-th data point. Here, $n$ is the dataset size, $d_0$ is the input dimension and $c$ is the number of classes. Moreover, we use $\boldsymbol{X} \in \mathbb{R}^{d_0 \times n}$ to stack all the input data into a matrix.

We consider an $L$-layer neural network of the form $f(\boldsymbol{\theta}; \boldsymbol{x})$, where $\boldsymbol{\theta}$ represents the model parameters, $\boldsymbol{x}$ is the input, and $f(\boldsymbol{\theta}; \boldsymbol{x}) \in \mathbb{R}^c$. Let the $\ell$-th layer feature (post-activation) of the network be $f^{(\ell)}(\boldsymbol{\theta}; \boldsymbol{x}) \in \mathbb{R}^{d_\ell}$, where $d_\ell$ is the dimension of the $\ell$-th layer ($0 \leq \ell \leq L$) and $d_L = c$. Note that

$f^{(0)}(\boldsymbol{\theta}; \boldsymbol{x}) = \boldsymbol{x}$ and $f^{(L)}(\boldsymbol{\theta}; \boldsymbol{x}) = f(\boldsymbol{\theta}; \boldsymbol{x})$. For an input data matrix $\boldsymbol{X}$, we also use $f(\boldsymbol{\theta}; \boldsymbol{X}) \in \mathbb{R}^{c \times n}$ and $f^{(\ell)}(\boldsymbol{\theta}; \boldsymbol{X}) \in \mathbb{R}^{d_\ell \times n}$ to denote the collection of the network outputs and features on all the datapoints, respectively. When $\boldsymbol{X}$ is clear from the context, we simply write $f^{(\ell)}(\boldsymbol{\theta}) = f^{(\ell)}(\boldsymbol{\theta}; \boldsymbol{X})$ and $f(\boldsymbol{\theta}) = f(\boldsymbol{\theta}; \boldsymbol{X})$. Unless otherwise specified, in this paper we consider models trained on a training set, and then all the investigations are evaluated on a test set.

We use $\mathrm{Err}_{\mathcal{D}}(\boldsymbol{\theta})$ to denote the classification error of the network $f(\boldsymbol{\theta}; \cdot)$ on the dataset $\mathcal{D}$.

**Linear Mode Connectivity (LMC).** We recall the notion of LMC in Definition 1.

**Definition 1** (**Linear Mode Connectivity**). *Given a test dataset $\mathcal{D}$ and two modes[3] $\boldsymbol{\theta}_A$ and $\boldsymbol{\theta}_B$ such that $\mathrm{Err}_{\mathcal{D}}(\boldsymbol{\theta}_A) \approx \mathrm{Err}_{\mathcal{D}}(\boldsymbol{\theta}_B)$, we say $\boldsymbol{\theta}_A$ and $\boldsymbol{\theta}_B$ are linearly connected if they satisfy*

$$\mathrm{Err}_{\mathcal{D}}(\alpha \boldsymbol{\theta}_A + (1 - \alpha)\boldsymbol{\theta}_B) \approx \mathrm{Err}_{\mathcal{D}}(\boldsymbol{\theta}_A), \qquad \forall \alpha \in [0, 1]. \tag{2}$$

As Definition 1 shows, $\boldsymbol{\theta}_A$ and $\boldsymbol{\theta}_B$ satisfy LMC if the error metric on the linear path connecting their weights is nearly constant. There are two known methods to obtain linearly connected modes, the *spawning method* [9, 7] and the *permutation method* [6, 1].

**Spawning Method.** We start from random initialization $\boldsymbol{\theta}^{(0)}$ and train the model for $k$ steps to obtain $\boldsymbol{\theta}^{(k)}$. Then we create two copies of $\boldsymbol{\theta}^{(k)}$ and continue training the two models separately using independent SGD randomnesses (mini-batch order and data augmentations) until convergence. By selecting a proper value of $k$ (usually a small fraction of the total training steps), we can obtain two linearly connected modes.

**Permutation Method.** Due to permutation symmetry, it is possible to permute the weights in a neural network appropriately while not changing the function being computed. Given two modes $\boldsymbol{\theta}_A$ and $\boldsymbol{\theta}_B$ which are independently trained (and not linearly connected), the permutation method aims to find a permutation $\pi$ such that the permuted mode $\boldsymbol{\theta}'_B = \pi(\boldsymbol{\theta}_B)$ is functionally equivalent to $\boldsymbol{\theta}_B$ and that $\boldsymbol{\theta}'_B$ and $\boldsymbol{\theta}_A$ are linearly connected. In other words, even if two modes are not linearly connected in the parameter space, they might still be linearly connected if permutation invariance is taken into account.

Among existing permutation methods, Git Re-Basin [1] is a representative one, which successfully achieved linear connectivity between two independently trained ResNet models on CIFAR-10. Specifically, a permutation $\pi$ that maintains functionally equivalent network can be formulated by a set of per-layer permutations $\pi = \{\boldsymbol{P}^{(\ell)}\}_{\ell=1}^{L-1}$ where $\boldsymbol{P}^{(\ell)} \in \mathbb{R}^{d_\ell \times d_\ell}$ is a permutation matrix. Ainsworth et al. [1] proposed two distinct matching objectives for aligning the neurons of independently trained models via permutation: *weight matching* and *activation matching*:

$$\text{Weight matching:} \quad \min_{\pi} \|\boldsymbol{\theta}_A - \pi(\boldsymbol{\theta}_B)\|^2. \tag{3}$$

$$\text{Activation matching:} \quad \min_{\pi} \sum_{\ell=1}^{L-1} \|\boldsymbol{H}_A^{(\ell)} - \boldsymbol{P}^{(\ell)} \boldsymbol{H}_B^{(\ell)}\|_F^2. \tag{4}$$

Here, $\boldsymbol{H}_A^{(\ell)} = f^{(\ell)}(\boldsymbol{\theta}_A; \boldsymbol{X})$ is the $\ell$-th layer feature matrix, and $\boldsymbol{H}_B^{(\ell)}$ is defined similarly.

**Main Experimental Setup.** In this paper, we use both the spawning method and the permutation method to obtain linearly connected modes. Following Frankle et al. [9], Ainsworth et al. [1], we perform our experiments on commonly used image classification datasets MNIST [18], CIFAR-10 [15], and Tiny-ImageNet[17], and with the standard network architectures ResNet-20/50 [12], VGG-16 [29], and MLP. We follow the same training procedures and hyper-parameters as in Frankle et al. [9], Ainsworth et al. [1]. Due to space limit, we defer some of the experimental results to the appendix. Notice that [1] increased the width of ResNet by 32 times in order to achieve zero barrier, and we also followed this setting in the experiments of the permutation method. The detailed settings and hyper-parameters are also described in Appendix B.1.

## 4 Layerwise Linear Feature Connectivity (LLFC)

In this section, we formally describe Layerwise Linear Feature Connectivity (LLFC) and provide empirical evidence of its consistent co-occurrence with LMC. We also show that LLFC applied to the last layer directly implies LMC.

---

[3]Following the terminology in literature, a *mode* refers to an optimal solution obtained at the end of training.

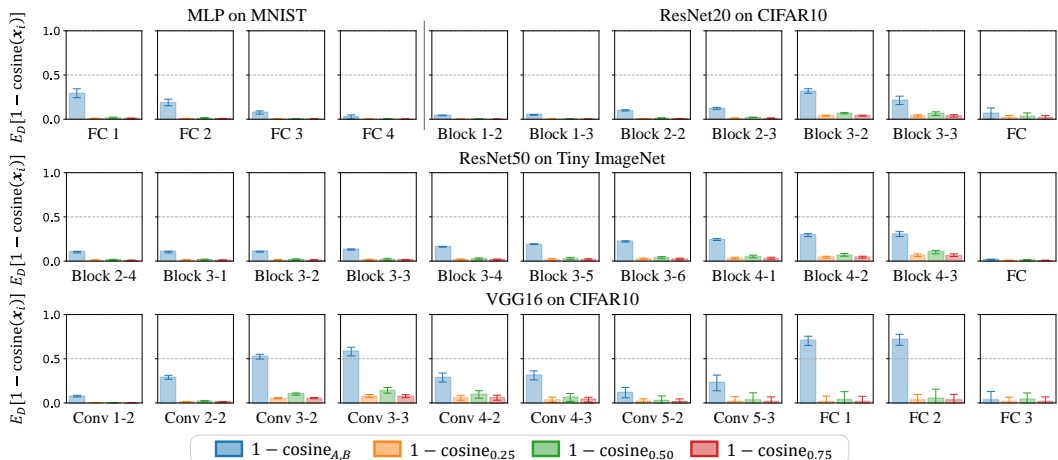

Figure 2: Comparison of $\mathbb{E}_{\mathcal{D}}[1 - \text{cosine}_\alpha(\boldsymbol{x}_i)]$ and $\mathbb{E}_{\mathcal{D}}[1 - \text{cosine}_{A,B}(\boldsymbol{x}_i)]$. The spawning method is used to obtain two linearly connected modes $\boldsymbol{\theta}_A$ and $\boldsymbol{\theta}_B$. Results are presented for different layers of various model architectures on different datasets, with $\alpha \in \{0.25, 0.5, 0.75\}$. Standard deviations across the dataset are reported by error bars. More results are in Appendix B.2.

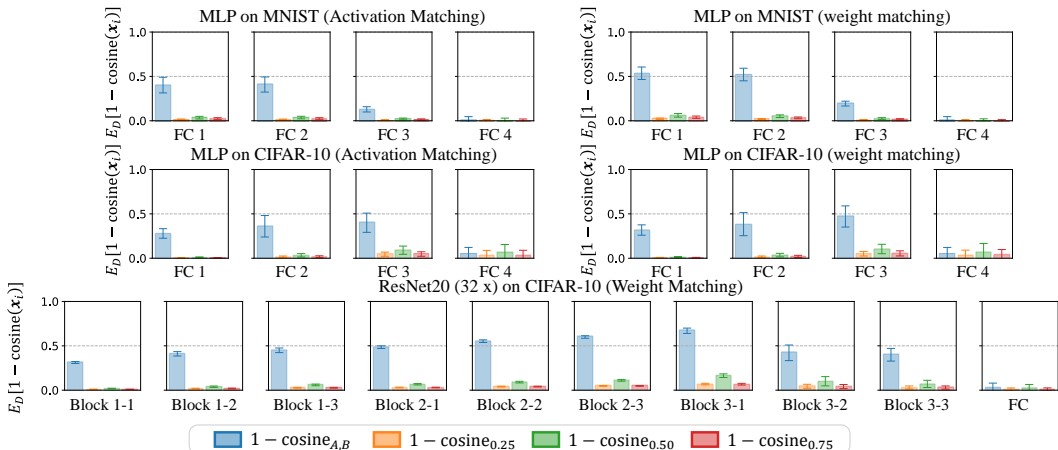

Figure 3: Comparison of $\mathbb{E}_{\mathcal{D}}[1 - \text{cosine}_\alpha(\boldsymbol{x}_i)]$ and $\mathbb{E}_{\mathcal{D}}[1 - \text{cosine}_{A,B}(\boldsymbol{x}_i)]$. The permutation method is used to obtain two linearly connected modes $\boldsymbol{\theta}_A$ and $\boldsymbol{\theta}_B$. Results are presented for different layers of various model architectures on different datasets, with $\alpha \in \{0.25, 0.5, 0.75\}$. Standard deviations across the dataset are reported by error bars. More results are in Appendix B.2.

**Definition 2 (Layerwise Linear Feature Connectivity).** *Given dataset $\mathcal{D}$ and two modes $\boldsymbol{\theta}_A$, $\boldsymbol{\theta}_B$ of an $L$-layer neural network $f$, the modes $\boldsymbol{\theta}_A$ and $\boldsymbol{\theta}_B$ are said to be layerwise linearly feature connected if they satisfy*

$$\forall \ell \in [L], \forall \alpha \in [0,1], \exists c > 0, s.t. \ cf^{(\ell)}(\alpha\boldsymbol{\theta}_A + (1-\alpha)\boldsymbol{\theta}_B) = \alpha f^{(\ell)}(\boldsymbol{\theta}_A) + (1-\alpha)f^{(\ell)}(\boldsymbol{\theta}_B). \tag{5}$$

LLFC states that the per-layer feature of the interpolated model $\boldsymbol{\theta}_\alpha = \alpha\boldsymbol{\theta}_A + (1-\alpha)\boldsymbol{\theta}_B$ has the same direction as the linear interpolation of the features of $\boldsymbol{\theta}_A$ and $\boldsymbol{\theta}_B$. This means that the feature map $f^{(\ell)}$ behaves similarly to a linear map (up to a scaling factor) on the line segment between $\boldsymbol{\theta}_A$ and $\boldsymbol{\theta}_B$, even though it is a nonlinear map globally.

**LLFC Co-occurs with LMC.** We now verify that LLFC consistently co-occurs with LMC across different architectures and datasets. We use the spawning method and the permutation method described in Section 3 to obtain linearly connected modes $\boldsymbol{\theta}_A$ and $\boldsymbol{\theta}_B$. On each data point $\boldsymbol{x}_i$ in the test set $\mathcal{D}$, we measure the cosine similarity between the feature of the interpolated model $\boldsymbol{\theta}_\alpha$ and

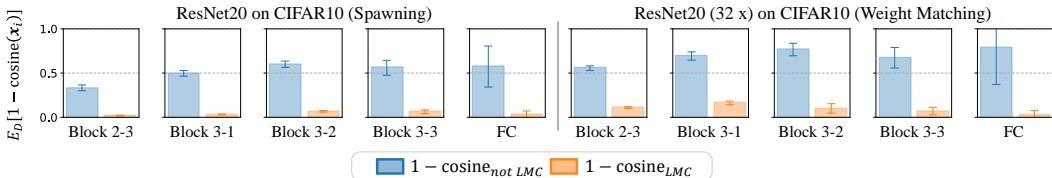

Figure 4: Comparison of $\mathbb{E}_{\mathcal{D}}[1 - \text{cosine}_{LMC}(\boldsymbol{x}_i)]$ and $\mathbb{E}_{\mathcal{D}}[1 - \text{cosine}_{not\ LMC}(\boldsymbol{x}_i)]$. Both the spawning and permutation methods are used to obtain two linearly connected modes. Standard deviations across the dataset are reported by error bars.

linear interpolations of the features of $\boldsymbol{\theta}_A$ and $\boldsymbol{\theta}_B$ in each layer $\ell$, as expressed as $\text{cosine}_\alpha(\boldsymbol{x}_i) = \cos[f^{(\ell)}(\alpha\boldsymbol{\theta}_A + (1-\alpha)\boldsymbol{\theta}_B; \boldsymbol{x}_i), \alpha f^{(\ell)}(\boldsymbol{\theta}_A; \boldsymbol{x}_i) + (1-\alpha)f^{(\ell)}(\boldsymbol{\theta}_B; \boldsymbol{x}_i)]$. We compare this to the baseline cosine similarity between the features of $\boldsymbol{\theta}_A$ and $\boldsymbol{\theta}_B$ in the corresponding layer, namely $\text{cosine}_{A,B}(\boldsymbol{x}_i) = \cos[f^{(\ell)}(\boldsymbol{\theta}_A; \boldsymbol{x}_i), f^{(\ell)}(\boldsymbol{\theta}_B; \boldsymbol{x}_i)]$. The results for the spawning method and the permutation method are presented in Figures 2 and 3, respectively. They show that the values of $\mathbb{E}_{\mathcal{D}}[1 - \text{cosine}_\alpha(\boldsymbol{x}_i)]$ are close to 0 across different layers, architectures, datasets, and different values of $\alpha$, which verifies the LLFC property. The presence of small error bars indicates consistent behavior for each data point. Moreover, the values of $\mathbb{E}_{\mathcal{D}}[1 - \text{cosine}_{A,B}(\boldsymbol{x}_i)]$ are not close to 0, which rules out the trivial case that $f^{(\ell)}(\boldsymbol{\theta}_A)$ and $f^{(\ell)}(\boldsymbol{\theta}_B)$ are already perfectly aligned.

To further verify, we also compare the values of $\mathbb{E}_{\mathcal{D}}[1 - \text{cosine}_\alpha(\boldsymbol{x}_i)]$ of two linearly connected modes with those of two modes that are independently trained (not satisfying LMC). We measure $\text{cosine}_{0.5}$ of the features of two modes that are linearly connected and two modes that are independently trained, denoted as $\text{cosine}_{LMC}$ and $\text{cosine}_{not\ LMC}$ correspondingly. In Figure 4, the values of $\mathbb{E}_{\mathcal{D}}[1 - \text{cosine}_{LMC}(\boldsymbol{x}_i)]$ are negligible compared to $\mathbb{E}_{\mathcal{D}}[1 - \text{cosine}_{not\ LMC}(\boldsymbol{x}_i)]$. The experimental results align with Figures 2 and 3 and thus we firmly verify the claim that LLFC co-occurs with LMC.

**LLFC Implies LMC.** Intuitively, LLFC is a much stronger characterization than LMC since it establishes a linearity property in the high-dimensional feature map in every layer, rather than just for the final error. Lemma 1 below formally establishes that LMC is a consequence of LLFC by applying LLFC on the output layer.

**Lemma 1** (Proof in Appendix A.1). *Suppose two modes $\boldsymbol{\theta}_A$, $\boldsymbol{\theta}_B$ satisfy LLFC on a dataset $\mathcal{D}$ and $\max\{\text{Err}_{\mathcal{D}}(\boldsymbol{\theta}_A), \text{Err}_{\mathcal{D}}(\boldsymbol{\theta}_B)\} \le \epsilon$, then we have*

$$\forall \alpha \in [0, 1], \text{Err}_{\mathcal{D}}(\alpha\boldsymbol{\theta}_A + (1-\alpha)\boldsymbol{\theta}_B) \le 2\epsilon. \tag{6}$$

In summary, we see that the LMC property, which was used to study the loss landscapes in the entire parameter space, extends to the internal features in almost all the layers. LLFC offers much richer structural properties than LMC. In Section 5, we will dig deeper into the contributing factors to LLFC and leverage the insights to gain new understanding of the spawning and the permutation methods.

## 5 Why Does LLFC Emerge?

We have seen that LLFC is a prevalent phenomenon that co-occurs with LMC, and it establishes a broader notion of linear connectivity than LMC. In this section, we investigate the root cause of LLFC, and identify two key conditions, *weak additivity for ReLU activations* and *commutativity*. We verify these conditions empirically and prove that they collectively imply LLFC. From there, we provide an explanation for the effectiveness of the permutation method, offering new insights into LMC.

### 5.1 Underlying Factors of LLFC

For convenience, we consider a multi-layer perceptron (MLP) in Section 5, though the results can be easily adapted to any feed-forward structure, e.g., a convolutional neural network[4] (CNN). For

---

[4]We also conduct experiments on CNNs. For a Conv layer, the forward propagation will be denoted as $\boldsymbol{W}\boldsymbol{H}$ similar to a linear layer. Typically, the weight $\boldsymbol{W}$ for a Conv layer has shape (# of output channels, # of input channels, height, width) and we reshape $\boldsymbol{W}$ to a matrix with dimensions (# of output channels, # of input channels × height × width).

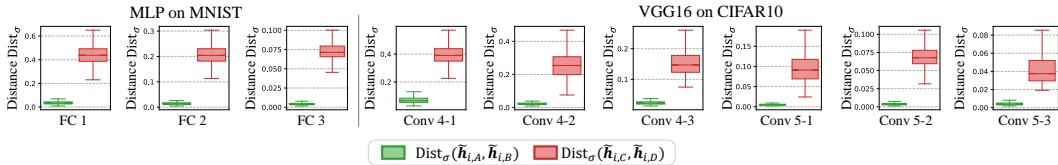

Figure 5: Comparison between the distribution of the normalized distance $\mathrm{Dist}_\sigma(\tilde{\boldsymbol{h}}_{i,A}, \tilde{\boldsymbol{h}}_{i,B})$ and $\mathrm{Dist}_\sigma(\tilde{\boldsymbol{h}}_{i,C}, \tilde{\boldsymbol{h}}_{i,D})$. Here, $\tilde{\boldsymbol{h}}_{i,A}$ and $\tilde{\boldsymbol{h}}_{i,B}$ are features of two linearly connected modes, i.e., $\boldsymbol{\theta}_A$ and $\boldsymbol{\theta}_B$ (founded by the spawning method). $\tilde{\boldsymbol{h}}_{i,C}$ and $\tilde{\boldsymbol{h}}_{i,D}$ comes from two modes that are independently trained. Results are presented for different layers of MLP on MNIST and VGG-16 on CIFAR-10.

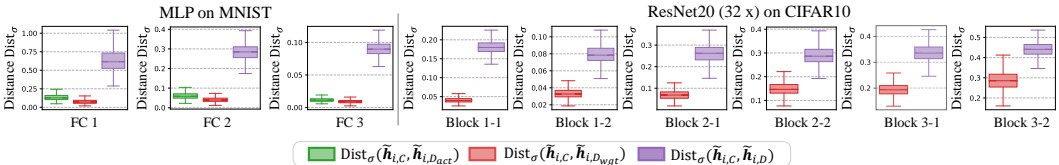

Figure 6: Comparison among the distribution of the normalized distance $\mathrm{Dist}_\sigma(\tilde{\boldsymbol{h}}_{i,C}, \tilde{\boldsymbol{h}}_{i,D_{act}})$, $\mathrm{Dist}_\sigma(\tilde{\boldsymbol{h}}_{i,C}, \tilde{\boldsymbol{h}}_{i,D_{wgt}})$ and $\mathrm{Dist}_\sigma(\tilde{\boldsymbol{h}}_{i,C}, \tilde{\boldsymbol{h}}_{i,D})$. Here, $\tilde{\boldsymbol{h}}_{i,C}$ and $\tilde{\boldsymbol{h}}_{i,D}$ are features of two modes that are independently trained, i.e., $\boldsymbol{\theta}_C$ and $\boldsymbol{\theta}_D$. $\tilde{\boldsymbol{h}}_{i,D_{act}}$ and $\tilde{\boldsymbol{h}}_{i,D_{wgt}}$ comes from $\boldsymbol{\theta}_{D_{act}}$ (permuted $\boldsymbol{\theta}_D$ using the activation matching) and $\boldsymbol{\theta}_{D_{wgt}}$ (permuted $\boldsymbol{\theta}_D$ using the weight matching), correspondingly. For ResNet-20, the values of $\mathrm{Dist}_\sigma$ are calculated in the first Conv layer of each block. Results are presented for different layers of MLP on MNIST and ResNet-20 on CIFAR-10.

an $L$-layer MLP $f$ with ReLU activation, the weight matrix in the $\ell$-th linear layer is denoted as $\boldsymbol{W}^{(\ell)} \in \mathbb{R}^{d_\ell \times d_{\ell-1}}$, and $\boldsymbol{b}^{(\ell)} \in \mathbb{R}^{d_\ell}$ is the bias in that layer. For a given input data matrix $\boldsymbol{X} \in \mathbb{R}^{d_0 \times n}$, denote the feature (post-activation) in the $\ell$-th layer as $\boldsymbol{H}^{(\ell)} = f^{(\ell)}(\boldsymbol{\theta}; \boldsymbol{X}) \in \mathbb{R}^{d_\ell \times n}$, and correspondingly pre-activation as $\tilde{\boldsymbol{H}}^{(\ell)} \in \mathbb{R}^{d_\ell \times n}$. The forward propagation in the $\ell$-th layer is:

$$\boldsymbol{H}^{(\ell)} = \sigma(\tilde{\boldsymbol{H}}^{(\ell)}), \quad \tilde{\boldsymbol{H}}^{(\ell)} = \boldsymbol{W}^{(\ell)}\boldsymbol{H}^{(\ell-1)} + \boldsymbol{b}^{(\ell)}\mathbf{1}_{d_\ell}^\top.$$

Here, $\sigma$ denotes the ReLU activation function, and $\mathbf{1}_{d_\ell} \in \mathbb{R}^{d_\ell}$ denotes the all-one vector. Additionally, we use $\boldsymbol{h}_i^{(\ell)}$ to denote the $i$-th row of $\boldsymbol{H}^{(\ell)}$, and $\tilde{\boldsymbol{h}}_i^{(\ell)}$ to denote the $i$-th row of $\tilde{\boldsymbol{H}}^{(\ell)}$, which correspond to the post- and pre-activations of the $i$-th input at layer $\ell$, respectively.

**Condition I: Weak Additivity for ReLU Activations.**[5]

**Definition 3** (**Weak Additivity for ReLU Activations**). *Given a dataset $\mathcal{D}$, two modes $\boldsymbol{\theta}_A$ and $\boldsymbol{\theta}_B$ are said to satisfy weak additivity for ReLU activations if*

$$\forall \ell \in [L], \forall \alpha \in [0,1], \quad \sigma(\alpha\tilde{\boldsymbol{H}}_A^{(\ell)} + (1-\alpha)\tilde{\boldsymbol{H}}_B^{(\ell)}) = \alpha\sigma(\tilde{\boldsymbol{H}}_A^{(\ell)}) + (1-\alpha)\sigma(\tilde{\boldsymbol{H}}_B^{(\ell)}). \quad (7)$$

Definition 3 requires the ReLU activation function to behave like a linear function for the pre-activations in each layer of the two networks. Although this cannot be true in general since ReLU is a nonlinear function, we verify it empirically for modes that satisfy LMC and LLFC.

We conduct experiments on various datasets and architectures to validate the weak additivity for ReLU activations. Specifically, given two modes $\boldsymbol{\theta}_A$ and $\boldsymbol{\theta}_B$ and a data point $\boldsymbol{x}_i$ in the test set $\mathcal{D}$, we compute the normalized distances between the left-hand side and the right-hand side of Equation (7) for each layer $\ell$, varying the values of $\alpha$. We denote the maximum distance across the range of $\alpha$ as $\mathrm{Dist}_\sigma(\tilde{\boldsymbol{h}}_{i,A}, \tilde{\boldsymbol{h}}_{i,B}) = \max_{\alpha \in [0,1]} \mathrm{dist}(\sigma\left(\alpha\tilde{\boldsymbol{h}}_{i,A} + (1-\alpha)\tilde{\boldsymbol{h}}_{i,B}\right), \alpha\sigma(\tilde{\boldsymbol{h}}_{i,A}) + (1-\alpha)\sigma(\tilde{\boldsymbol{h}}_{i,B}))$, where $\mathrm{dist}(\boldsymbol{x}, \boldsymbol{y}) := \|\boldsymbol{x} - \boldsymbol{y}\|^2/(\|\boldsymbol{x}\| \cdot \|\boldsymbol{y}\|)$. To validate the weak additivity condition, we compare the values of $\mathrm{Dist}_\sigma$ of two modes that are linearly connected with those of two modes that are independently trained.

---

[5]Weak additivity has no relation to stable neurons [28]. Stable neuron is defined as one whose output is the constant value zero or the pre-activation output on all inputs, which is a property concerning a single network. On the other hand, weak additivity concerns a relation between two networks.

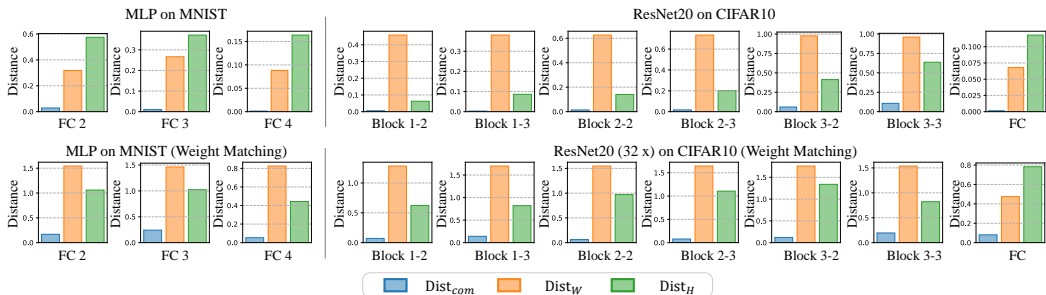

Figure 7: Comparison of $\text{Dist}_{com}$, $\text{Dist}_W$, and $\text{Dist}_H$. In the first row, the spawning method is used to acquire modes that satisfy LLFC, whereas the permutation method is used for the second row. For ResNet-20, $\text{Dist}_{com}$, $\text{Dist}_W$ are calculated in the first Conv layer of each block. The results are presented for different layers of MLP on the MNIST and ResNet-20 on the CIFAR-10. More results are in Appendix B.3.

Both spawning and permutation methods are shown to demonstrate the weak additivity condition. For spawning method, we first obtain two linearly connected modes, i.e, $\boldsymbol{\theta}_A$ and $\boldsymbol{\theta}_B$, and then two independently trained modes, i.e, $\boldsymbol{\theta}_C$ and $\boldsymbol{\theta}_D$ (not satisfying LMC/LLFC). We compare the values of $\text{Dist}_\sigma$ of $\boldsymbol{\theta}_A$ and $\boldsymbol{\theta}_B$ with those of $\boldsymbol{\theta}_C$ and $\boldsymbol{\theta}_D$. In Figure 5, we observe that across different datasets and model architectures, at different layers, $\text{Dist}_\sigma(\tilde{\boldsymbol{h}}_{i,A}, \tilde{\boldsymbol{h}}_{i,B})$ are negligible (and much smaller than the baseline $\text{Dist}_\sigma(\tilde{\boldsymbol{h}}_{i,C}, \tilde{\boldsymbol{h}}_{i,D})$). For permutation methods, given two independently trained modes, i.e., $\boldsymbol{\theta}_C$ and $\boldsymbol{\theta}_D$, we permute the $\boldsymbol{\theta}_D$ such that the permuted $\pi(\boldsymbol{\theta}_D)$ are linearly connected with $\boldsymbol{\theta}_C$. Both activation matching and weight matching are used and the permuted $\pi(\boldsymbol{\theta}_D)$ are denoted as $\boldsymbol{\theta}_{D_{act}}$ and $\boldsymbol{\theta}_{D_{wgt}}$ respectively. In Figure 6, the values of $\text{Dist}_\sigma(\tilde{\boldsymbol{h}}_{i,C}, \tilde{\boldsymbol{h}}_{i,D_{act}})$ and $\text{Dist}_\sigma(\tilde{\boldsymbol{h}}_{i,C}, \tilde{\boldsymbol{h}}_{i,D_{wgt}})$ are close to zero compared to $\text{Dist}_\sigma(\tilde{\boldsymbol{h}}_{i,C}, \tilde{\boldsymbol{h}}_{i,D})$. Therefore, we verify the weak additivity condition for both spawning and permutation methods.

**Condition II: Commutativity.**

**Definition 4** (**Commutativity**). *Given a dataset* $\mathcal{D}$, *two modes* $\boldsymbol{\theta}_A$ *and* $\boldsymbol{\theta}_B$ *are said to satisfy commutativity if*

$$\forall \ell \in [L],\ \boldsymbol{W}_A^{(\ell)}\boldsymbol{H}_A^{(\ell-1)} + \boldsymbol{W}_B^{(\ell)}\boldsymbol{H}_B^{(\ell-1)} = \boldsymbol{W}_A^{(\ell)}\boldsymbol{H}_B^{(\ell-1)} + \boldsymbol{W}_B^{(\ell)}\boldsymbol{H}_A^{(\ell-1)}. \tag{8}$$

Commutativity depicts that the next-layer linear transformations applied to the internal features of two neural networks can be interchanged. This property is crucial for improving our understanding of LMC and LLFC. In Section 5.2, we will use the commutativity property to provide new insights into the permutation method.

We conduct experiments on various datasets and model architectures to verify the commutativity property for modes that satisfy LLFC. Specifically, for a given dataset $\mathcal{D}$ and two modes $\boldsymbol{\theta}_A$ and $\boldsymbol{\theta}_B$ that satisfy LLFC, we compute the normalized distance between the left-hand side and the right-hand side of Equation (8), denoted as $\text{Dist}_{com} = \text{dist}\left(\text{vec}(\boldsymbol{W}_A^{(\ell)}\boldsymbol{H}_A^{(\ell-1)} + \boldsymbol{W}_B^{(\ell)}\boldsymbol{H}_B^{(\ell-1)}), \text{vec}(\boldsymbol{W}_A^{(\ell)}\boldsymbol{H}_B^{(\ell-1)} + \boldsymbol{W}_B^{(\ell)}\boldsymbol{H}_A^{(\ell-1)})\right)$. Furthermore, we compare $\text{Dist}_{com}$ with the normalized distance between the weight matrices of the current layer $\ell$, denoted as $\text{Dist}_W$, and the normalized distances between the post-activations of the previous layer $\ell - 1$, denoted as $\text{Dist}_H$. These distances are expressed as $\text{Dist}_W = \text{dist}\left(\text{vec}(\boldsymbol{W}_A^{(\ell)}), \text{vec}(\boldsymbol{W}_B^{(\ell)})\right)$ and $\text{Dist}_H = \text{dist}\left(\text{vec}(\boldsymbol{H}_A^{(\ell-1)}), \text{vec}(\boldsymbol{H}_B^{(\ell-1)})\right)$, respectively. Figure 7 shows that for both spawning and permutation methods, $\text{Dist}_{com}$ is negligible compared with $\text{Dist}_W$ and $\text{Dist}_H$, which confirms the commutativity condition. Note that we also rule out the trivial case where either the weight matrices or the post-activations in the two networks are already perfectly aligned, as weights and post-activations often differ significantly. We also add more baseline experiments for comparison (see Appendix B.3 for more results).

Additionally, we note that commutativity has a similarity to model stitching [19, 3]. In Appendix B.4, we show that a stronger form of model stitching (without an additional trainable layer) works for two networks that satisfy LLFC.

**Conditions I and II Imply LLFC.** Theorem 1 below shows that weak additivity for ReLU activations (Definition 3) and commutativity (Definition 4) imply LLFC.

**Theorem 1** (Proof in Appendix A.2). *Given a dataset $\mathcal{D}$, if two modes $\boldsymbol{\theta}_A$ and $\boldsymbol{\theta}_B$ satisfy weak additivity for ReLU activations (Definition 3) and commutativity (Definition 4), then*

$$\forall \alpha \in [0,1], \forall \ell \in [L],\ f^{(\ell)}\left(\alpha \boldsymbol{\theta}_A + (1-\alpha)\boldsymbol{\theta}_B\right) = \alpha f^{(\ell)}(\boldsymbol{\theta}_A) + (1-\alpha) f^{(\ell)}(\boldsymbol{\theta}_B).$$

Note that the definition of LLFC (Definition 2) allows a scaling factor $c$, while Theorem 1 establishes a stronger version of LLFC where $c = 1$. We attribute this inconsistency to the accumulation of errors[6] in weak additivity and commutativity conditions, since they are only approximated satisfied in practice. Yet in most cases, we observe that $c$ is close to 1 (see Appendix B.2 for more results).

## 5.2 Justification of the Permutation Methods

In this subsection, we provide a justification of the permutation methods in Git Re-Basin [1]: weight matching (3) and activation matching (4). Recall from Section 3 that given two modes $\boldsymbol{\theta}_A$ and $\boldsymbol{\theta}_B$, the permutation method aims to find a permutation $\pi = \{\boldsymbol{P}^{(\ell)}\}_{\ell=1}^{L-1}$ such that the permuted $\boldsymbol{\theta}'_B = \pi(\boldsymbol{\theta}_B)$ and $\boldsymbol{\theta}_A$ are linearly connected, where $\boldsymbol{P}^{(\ell)}$ is a permutation matrix applied to the $\ell$-th layer feature. Concretely, with a permutation $\pi$, we can formulate $\boldsymbol{W'}_B^{(\ell)}$ and $\boldsymbol{H'}_B^{(\ell)}$ of $\boldsymbol{\theta}'_B$ in each layer $\ell$ as $\boldsymbol{W'}_B^{(\ell)} = \boldsymbol{P}^{(\ell)}\boldsymbol{W}_B^{(\ell)}\boldsymbol{P}^{(\ell-1)\top}$ and $\boldsymbol{H'}_B^{(\ell)} = \boldsymbol{P}^{(\ell)}\boldsymbol{H}_B^{(\ell)}$ [1].[7] In Section 5.1, we have identified the commutativity property as a key factor contributing to LLFC. The commutativity property (8) between $\boldsymbol{\theta}_A$ and $\boldsymbol{\theta}'_B$ can be written as

$$\left(\boldsymbol{W}_A^{(\ell)} - \boldsymbol{W'}_B^{(\ell)}\right)\left(\boldsymbol{H}_A^{(\ell-1)} - \boldsymbol{H'}_B^{(\ell-1)}\right) = \boldsymbol{0},$$

or

$$\left(\boldsymbol{W}_A^{(\ell)} - \boldsymbol{P}^{(\ell)}\boldsymbol{W}_B^{(\ell)}\boldsymbol{P}^{(\ell-1)\top}\right)\left(\boldsymbol{H}_A^{(\ell-1)} - \boldsymbol{P}^{(\ell-1)}\boldsymbol{H}_B^{(\ell-1)}\right) = \boldsymbol{0}. \tag{9}$$

We note that the weight matching (3) and activation matching (4) objectives can be written as $\min_\pi \sum_{\ell=1}^{L}\left\|\boldsymbol{W}_A^{(\ell)} - \boldsymbol{P}^{(\ell)}\boldsymbol{W}_B^{(\ell)}\boldsymbol{P}^{(\ell-1)\top}\right\|_F^2$ and $\min_\pi \sum_{\ell=1}^{L-1}\left\|\boldsymbol{H}_A^{(\ell)} - \boldsymbol{P}^{(\ell)}\boldsymbol{H}_B^{(\ell)}\right\|_F^2$, respectively, which directly correspond to the two factors in Equation (9). Therefore, we can interpret Git Re-Basin as a means to ensure commutativity. Extended discussion on Git Re-basin [1] can be found in Appendix B.5.

## 6 Conclusion and Discussion

We identified Layerwise Linear Feature Connectivity (LLFC) as a prevalent phenomenon that co-occurs with Linear Mode Connectivity (LMC). By investigating the underlying contributing factors to LLFC, we obtained novel insights into the existing permutation methods that give rise to LMC. The consistent co-occurrence of LMC and LLFC suggests that LLFC may play an important role if we want to understand LMC in full. Since the LLFC phenomenon suggests that averaging weights is roughly equivalent to averaging features, a natural future direction is to study feature averaging methods and investigate whether averaging leads to better features.

We note that our current experiments mainly focus on image classification tasks, though aligning with existing literature on LMC. We leave the exploration of empirical evidence beyond image classification as future direction. We also note that our Theorem 1 predicts LLFC in an ideal case, while in practice, a scaling factor $c$ is introduced to Definition 2 to better describe the experimental results. Realistic theorems and definitions (approximated version) are defered to future research.

Finally, we leave the question if it is possible to find a permutation directly enforcing the commutativity property (9). Minimizing $\|(\boldsymbol{W}_A^{(\ell)} - \boldsymbol{P}^{(\ell)}\boldsymbol{W}_B^{(\ell)}\boldsymbol{P}^{(\ell-1)\top})(\boldsymbol{H}_A^{(\ell-1)} - \boldsymbol{P}^{(\ell-1)}\boldsymbol{H}_B^{(\ell-1)})\|_F$ entails solving Quadratic Assignment Problems (QAPs) (See Appendix A.3 for derivation), which are known NP-hard. Solving QAPs calls for efficient techniques especially seeing the progress in learning to solve QAPs [37, 34].

---

[6]we find that employing a scaling factor $c$ enables a much better description of the practical behavior than other choices (e.g. an additive error term).

[7]Note that both $\boldsymbol{P}^{(0)}$ and $\boldsymbol{P}^{(L)}$ are identity matrices.

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

# A   Proofs of Theorems and Derivations

## A.1   Proof of Lemma 1

In this section, we prove Lemma 1 in the main paper. This lemma indicates that we can directly imply Linear Mode Connectivity (LMC, see Definition 1) from Layerwise Linear Feature Connectivity (LLFC, see Definition 2) applied to last layer.

**Definition 1** (**Linear Mode Connectivity**). *Given a test dataset $\mathcal{D}$ and two modes $\boldsymbol{\theta}_A$ and $\boldsymbol{\theta}_B$ such that $\mathrm{Err}_{\mathcal{D}}(\boldsymbol{\theta}_A) \approx \mathrm{Err}_{\mathcal{D}}(\boldsymbol{\theta}_B)$, we say $\boldsymbol{\theta}_A$ and $\boldsymbol{\theta}_B$ are linearly connected if they satisfy*

$$\mathrm{Err}_{\mathcal{D}}(\alpha\boldsymbol{\theta}_A + (1-\alpha)\boldsymbol{\theta}_B) \approx \mathrm{Err}_{\mathcal{D}}(\boldsymbol{\theta}_A), \qquad \forall \alpha \in [0,1].$$

**Definition 2** (**Layerwise Linear Feature Connectivity**). *Given dataset $\mathcal{D}$ and two modes $\boldsymbol{\theta}_A$, $\boldsymbol{\theta}_B$ of an $L$-layer neural network $f$, the modes $\boldsymbol{\theta}_A$ and $\boldsymbol{\theta}_B$ are said to be layerwise linearly feature connected if they satisfy*

$$\forall \ell \in [L], \forall \alpha \in [0,1], \exists c > 0, s.t. \ cf^{(\ell)}(\alpha\boldsymbol{\theta}_A + (1-\alpha)\boldsymbol{\theta}_B) = \alpha f^{(\ell)}(\boldsymbol{\theta}_A) + (1-\alpha)f^{(\ell)}(\boldsymbol{\theta}_B).$$

**Lemma 1.** *Suppose two modes $\boldsymbol{\theta}_A$, $\boldsymbol{\theta}_B$ satisfy LLFC on a dataset $\mathcal{D}$ and*

$$\max\{\mathrm{Err}_{\mathcal{D}}(\boldsymbol{\theta}_A), \mathrm{Err}_{\mathcal{D}}(\boldsymbol{\theta}_B)\} \leq \epsilon,$$

*then we have*

$$\forall \alpha \in [0,1], \mathrm{Err}_{\mathcal{D}}(\alpha\boldsymbol{\theta}_A + (1-\alpha)\boldsymbol{\theta}_B) \leq 2\epsilon.$$

*Proof.* Note that the classification depends on the relative order of the entries in the output of the final layer. As a consequence, for each data point in the dataset $\mathcal{D}$, the linear interpolation of the outputs of the models makes the correct classification if both models make the correct classification. Therefore, only if one of the model makes the incorrect classification, the linear interpolation of the outputs of the models would possibly make the incorrect classification, i.e,

$$\mathrm{Err}_{\mathcal{D}}(\alpha f(\boldsymbol{\theta}_A) + (1-\alpha)f(\boldsymbol{\theta}_B)) \leq \mathrm{Err}_{\mathcal{D}}(\boldsymbol{\theta}_A) + \mathrm{Err}_{\mathcal{D}}(\boldsymbol{\theta}_B).$$

Since $\boldsymbol{\theta}_A$ and $\boldsymbol{\theta}_B$ satisfy LLFC, then at last layer we have

$$f(\alpha\boldsymbol{\theta}_A + (1-\alpha)\boldsymbol{\theta}_B) = \alpha f(\boldsymbol{\theta}_A) + (1-\alpha)f(\boldsymbol{\theta}_B),$$

then have

$$\mathrm{Err}_{\mathcal{D}}(\alpha\boldsymbol{\theta}_A + (1-\alpha)\boldsymbol{\theta}_B) \leq \mathrm{Err}_{\mathcal{D}}(\boldsymbol{\theta}_A) + \mathrm{Err}_{\mathcal{D}}(\boldsymbol{\theta}_B).$$

According to the condition that

$$\max\{\mathrm{Err}_{\mathcal{D}}(\boldsymbol{\theta}_A), \mathrm{Err}_{\mathcal{D}}(\boldsymbol{\theta}_B)\} \leq \epsilon,$$

which indicates

$$\mathrm{Err}_{\mathcal{D}}(\alpha\boldsymbol{\theta}_A + (1-\alpha)\boldsymbol{\theta}_B) \leq 2\epsilon,$$

and this finishes the proof. □

## A.2   Proof of Theorem 1

In this section, we prove Theorem 1 in the main paper. Theorem 1 indicates that we can derive LLFC from two simple conditions: weak additivity for ReLU activations (Definition 3) and commutativity (Definition 4). Note that though we consider a multi-layer perceptron (MLP) for convenience, our proof and results can be easily adopted to any feed-forward structure, e.g., a convolutional neural network (CNN).

**Definition 3** (**Weak Additivity for ReLU Activations**). *Given a dataset $\mathcal{D}$, two modes $\boldsymbol{\theta}_A$ and $\boldsymbol{\theta}_B$ are said to satisfy weak additivity for ReLU activations if*

$$\forall \ell \in [L], \quad \sigma\left(\tilde{\boldsymbol{H}}_A^{(\ell)} + \tilde{\boldsymbol{H}}_B^{(\ell)}\right) = \sigma\left(\tilde{\boldsymbol{H}}_A^{(\ell)}\right) + \sigma\left(\tilde{\boldsymbol{H}}_B^{(\ell)}\right).$$

**Definition 4** (**Commutativity**). *Given a dataset $\mathcal{D}$, two modes $\boldsymbol{\theta}_A$ and $\boldsymbol{\theta}_B$ are said to satisfy commutativity if*

$$\forall \ell \in [L], \boldsymbol{W}_A^{(\ell)} \boldsymbol{H}_A^{(\ell-1)} + \boldsymbol{W}_B^{(\ell)} \boldsymbol{H}_B^{(\ell-1)} = \boldsymbol{W}_A^{(\ell)} \boldsymbol{H}_B^{(\ell-1)} + \boldsymbol{W}_B^{(\ell)} \boldsymbol{H}_A^{(\ell-1)}.$$

**Theorem 1.** *Given a dataset $\mathcal{D}$, if two modes $\boldsymbol{\theta}_A$ and $\boldsymbol{\theta}_B$ satisfy weak additivity for ReLU activations (Definition 3) and commutativity (Definition 4), then*

$$\forall \alpha \in [0,1], \forall \ell \in [L], \ f^{(\ell)}\left(\alpha\boldsymbol{\theta}_A + (1-\alpha)\boldsymbol{\theta}_B\right) = \alpha f^{(\ell)}\left(\boldsymbol{\theta}_A\right) + (1-\alpha) f^{(\ell)}\left(\boldsymbol{\theta}_B\right).$$

*Proof.* Before delving into the proof, let us denote the forward propagation in each layer $\ell$ by

$$\tilde{g}^{(\ell)}\left(\boldsymbol{\theta}; \boldsymbol{H}^{(\ell-1)}\right) = \boldsymbol{W}^{(\ell)} \boldsymbol{H}^{(\ell-1)} + \boldsymbol{b}^{(\ell)} \mathbf{1}_{d_\ell}^\top$$

$$g^{(\ell)}\left(\boldsymbol{\theta}; \boldsymbol{H}^{(\ell-1)}\right) = \sigma\left(\tilde{g}^{(\ell)}\left(\boldsymbol{\theta}; \boldsymbol{H}^{(\ell-1)}\right)\right) = \boldsymbol{H}^{(\ell)}$$

Given $\boldsymbol{\theta}_A$ and $\boldsymbol{\theta}_B$ that satisfy the commutativity property, then $\forall \ell \in [L]$ and $\forall \alpha \in [0,1]$, we have

$$\boldsymbol{W}_A^{(\ell)} \boldsymbol{H}_A^{(\ell-1)} + \boldsymbol{W}_B^{(\ell)} \boldsymbol{H}_B^{(\ell-1)} = \boldsymbol{W}_A^{(\ell)} \boldsymbol{H}_B^{(\ell-1)} + \boldsymbol{W}_B^{(\ell)} \boldsymbol{H}_A^{(\ell-1)}$$

$$\tilde{g}^{(\ell)}\left(\boldsymbol{\theta}_A; \boldsymbol{H}_A^{(\ell-1)}\right) + \tilde{g}^{(\ell)}\left(\boldsymbol{\theta}_B; \boldsymbol{H}_B^{(\ell-1)}\right) = \tilde{g}^{(\ell)}\left(\boldsymbol{\theta}_A; \boldsymbol{H}_B^{(\ell-1)}\right) + \tilde{g}^{(\ell)}\left(\boldsymbol{\theta}_B; \boldsymbol{H}_A^{(\ell-1)}\right)$$

$$\alpha(1-\alpha)\left(\tilde{g}^{(\ell)}\left(\boldsymbol{\theta}_A; \boldsymbol{H}_A^{(\ell-1)}\right) + \tilde{g}^{(\ell)}\left(\boldsymbol{\theta}_B; \boldsymbol{H}_B^{(\ell-1)}\right)\right) = \alpha(1-\alpha)\left(\tilde{g}^{(\ell)}\left(\boldsymbol{\theta}_A; \boldsymbol{H}_B^{(\ell-1)}\right) + \tilde{g}^{(\ell)}\left(\boldsymbol{\theta}_B; \boldsymbol{H}_A^{(\ell-1)}\right)\right)$$

$$\alpha\tilde{g}^{(\ell)}\left(\boldsymbol{\theta}_A; \boldsymbol{H}_A^{(\ell-1)}\right) + (1-\alpha)\tilde{g}^{(\ell)}\left(\boldsymbol{\theta}_B; \boldsymbol{H}_B^{(\ell-1)}\right) = \alpha^2 \tilde{g}^{(\ell)}\left(\boldsymbol{\theta}_A; \boldsymbol{H}_A^{(\ell-1)}\right) + (1-\alpha)^2 \tilde{g}^{(\ell)}\left(\boldsymbol{\theta}_B; \boldsymbol{H}_B^{(\ell-1)}\right)$$
$$+ \alpha(1-\alpha)\left(\tilde{g}^{(\ell)}\left(\boldsymbol{\theta}_A; \boldsymbol{H}_B^{(\ell-1)}\right) + \tilde{g}^{(\ell)}\left(\boldsymbol{\theta}_B; \boldsymbol{H}_A^{(\ell-1)}\right)\right)$$

Additionally, we can easily verify that

$$\tilde{g}^{(\ell)}\left(\alpha\boldsymbol{\theta}_A + (1-\alpha)\boldsymbol{\theta}_B; \boldsymbol{H}^{(\ell)}\right) = \alpha\tilde{g}^{(\ell)}\left(\boldsymbol{\theta}_A; \boldsymbol{H}^{(\ell)}\right) + (1-\alpha)\tilde{g}^{(\ell)}\left(\boldsymbol{\theta}_B; \boldsymbol{H}^{(\ell)}\right)$$

$$\tilde{g}^{(\ell)}\left(\boldsymbol{\theta}; \alpha\boldsymbol{H}_A^{(\ell)} + (1-\alpha)\boldsymbol{H}_B^{(\ell)}\right) = \alpha\tilde{g}^{(\ell)}\left(\boldsymbol{\theta}; \boldsymbol{H}_A^{(\ell)}\right) + (1-\alpha)\tilde{g}^{(\ell)}\left(\boldsymbol{\theta}; \boldsymbol{H}_B^{(\ell)}\right)$$

Subsequently,

$$\alpha\tilde{g}^{(\ell)}\left(\boldsymbol{\theta}_A; \boldsymbol{H}_A^{(\ell-1)}\right) + (1-\alpha)\tilde{g}^{(\ell)}\left(\boldsymbol{\theta}_B; \boldsymbol{H}_B^{(\ell-1)}\right) = \alpha\tilde{g}^{(\ell)}\left(\alpha\boldsymbol{\theta}_A + (1-\alpha)\boldsymbol{\theta}_B; \boldsymbol{H}_A^{(\ell-1)}\right)$$
$$+ (1-\alpha)\tilde{g}^{(\ell)}\left(\alpha\boldsymbol{\theta}_A + (1-\alpha)\boldsymbol{\theta}_B; \boldsymbol{H}_B^{(\ell-1)}\right)$$
$$= \tilde{g}^{(\ell)}\left(\alpha\boldsymbol{\theta}_A + (1-\alpha)\boldsymbol{\theta}_B; \alpha\boldsymbol{H}_A^{(\ell-1)} + (1-\alpha)\boldsymbol{H}_B^{(\ell-1)}\right).$$

Given the weak additivity for ReLU activation is satisfied for $\boldsymbol{\theta}_A$ and $\boldsymbol{\theta}_B$, then we have

$$\sigma\left(\alpha\tilde{g}^{(\ell)}\left(\boldsymbol{\theta}_A; \boldsymbol{H}_A^{(\ell-1)}\right) + (1-\alpha)\tilde{g}^{(\ell)}\left(\boldsymbol{\theta}_B; \boldsymbol{H}_B^{(\ell-1)}\right)\right) = \sigma\left(\tilde{g}^{(\ell)}\left(\alpha\boldsymbol{\theta}_A + (1-\alpha)\boldsymbol{\theta}_B; \alpha\boldsymbol{H}_A^{(\ell-1)} + (1-\alpha)\boldsymbol{H}_B^{(\ell-1)}\right)\right)$$

$$\alpha g^{(\ell)}\left(\boldsymbol{\theta}_A; \boldsymbol{H}_A^{(\ell-1)}\right) + (1-\alpha) g^{(\ell)}\left(\boldsymbol{\theta}_B; \boldsymbol{H}_B^{(\ell-1)}\right) = g^{(\ell)}\left(\alpha\boldsymbol{\theta}_A + (1-\alpha)\boldsymbol{\theta}_B; \alpha\boldsymbol{H}_A^{(\ell-1)} + (1-\alpha)\boldsymbol{H}_B^{(\ell-1)}\right)$$

To conclude, $\forall \ell \in [L]$ and $\forall \alpha \in [0,1]$, we have

$$\alpha\boldsymbol{H}_A^{(\ell)} + (1-\alpha)\boldsymbol{H}_B^{(\ell)} = g^{(\ell)}\left(\alpha\boldsymbol{\theta}_A + (1-\alpha)\boldsymbol{\theta}_B; \alpha\boldsymbol{H}_A^{(\ell-1)} + (1-\alpha)\boldsymbol{H}_B^{(\ell-1)}\right) \qquad (10)$$

For the right hand side of Equation (10), recursively, we can have

$$g^{(\ell)}\left(\alpha\boldsymbol{\theta}_A + (1-\alpha)\boldsymbol{\theta}_B; \alpha\boldsymbol{H}_A^{(\ell-1)} + (1-\alpha)\boldsymbol{H}_B^{(\ell-1)}\right)$$
$$= g^{(\ell)}\left(\alpha\boldsymbol{\theta}_A + (1-\alpha)\boldsymbol{\theta}_B; g^{(\ell-1)}\left(\alpha\boldsymbol{\theta}_A + (1-\alpha)\boldsymbol{\theta}_B; \alpha\boldsymbol{H}_A^{(\ell-2)} + (1-\alpha)\boldsymbol{H}_B^{(\ell-2)}\right)\right)$$
$$= \left(g^{(\ell)} \circ g^{(\ell-1)}\right)\left(\alpha\boldsymbol{\theta}_A + (1-\alpha)\boldsymbol{\theta}_B; \alpha\boldsymbol{H}_A^{(\ell-2)} + (1-\alpha)\boldsymbol{H}_B^{(\ell-2)}\right)$$
$$= \cdots$$
$$= \left(g^{(\ell)} \circ g^{(\ell-1)} \cdots \circ g^{(1)}\right)\left(\alpha\boldsymbol{\theta}_A + (1-\alpha)\boldsymbol{\theta}_B; \boldsymbol{X}\right)$$
$$= f^{(\ell)}\left(\alpha\boldsymbol{\theta}_A + (1-\alpha)\boldsymbol{\theta}_B; \boldsymbol{X}\right),$$

where $\boldsymbol{X}$ denotes the input data matrix.

Recall we denote $\boldsymbol{H}^{(\ell)} = f^{(\ell)}\left(\boldsymbol{\theta}; \boldsymbol{X}\right)$ which indicates

$$\alpha f^{(\ell)}\left(\boldsymbol{\theta}_A; \boldsymbol{X}\right) + \left(1 - \alpha\right) f^{(\ell)}\left(\boldsymbol{\theta}_B; \boldsymbol{X}\right) = f^{(\ell)}\left(\alpha\boldsymbol{\theta}_A + \left(1 - \alpha\right)\boldsymbol{\theta}_B; \boldsymbol{X}\right),$$

and this finishes the proof. $\qquad\square$

### A.3 Derivation of Quadratic Assignment Problem

In this section, we aim to show that minimizing $\sum_{\ell=1}^{L}\left\|\left(\boldsymbol{W}_A^{(\ell)} - \boldsymbol{P}^{(\ell)}\boldsymbol{W}_B^{(\ell)}\boldsymbol{P}^{(\ell-1)^\top}\right)\left(\boldsymbol{H}_A^{(\ell-1)} - \boldsymbol{P}^{(\ell-1)}\boldsymbol{H}_B^{(\ell-1)}\right)\right\|_F^2$ includes solving Quadratic Assignment Problems (QAPs), known to be NP-hard.

$$\underset{\pi=\{\boldsymbol{P}^{(\ell)}\}}{\arg\min} \sum_{\ell=1}^{L}\left\|\left(\boldsymbol{W}_A^{(\ell)} - \boldsymbol{P}^{(\ell)}\boldsymbol{W}_B^{(\ell)}\boldsymbol{P}^{(\ell-1)^\top}\right)\left(\boldsymbol{H}_A^{(\ell-1)} - \boldsymbol{P}^{(\ell-1)}\boldsymbol{H}_B^{(\ell-1)}\right)\right\|_F^2$$

$$\Longleftrightarrow \underset{\pi=\{\boldsymbol{P}^{(\ell)}\}}{\arg\min} \sum_{\ell=1}^{L}\left\|\boldsymbol{W}_A^{(\ell)}\boldsymbol{H}_A^{(\ell-1)} - \boldsymbol{P}^{(\ell)}\boldsymbol{W}_B^{(\ell)}\boldsymbol{P}^{(\ell-1)^\top}\boldsymbol{H}_A^{(\ell-1)} - \boldsymbol{W}_A^{(\ell)}\boldsymbol{P}^{(\ell-1)}\boldsymbol{H}_B^{(\ell-1)} + \boldsymbol{P}^{(\ell)}\boldsymbol{W}_B^{(\ell)}\boldsymbol{H}_B^{(\ell-1)}\right\|_F^2$$

$$\Longleftrightarrow \underset{\pi=\{\boldsymbol{P}^{(\ell)}\}}{\arg\min} \sum_{\ell=1}^{L}\left(\left\|\boldsymbol{W}_A^{(\ell)}\boldsymbol{H}_A^{(\ell-1)} - \boldsymbol{W}_A^{(\ell)}\boldsymbol{P}^{(\ell-1)}\boldsymbol{H}_B^{(\ell-1)}\right\|_F^2 + \left\|\boldsymbol{P}^{(\ell)}\boldsymbol{W}_B^{(\ell)}\boldsymbol{P}^{(\ell-1)^\top}\boldsymbol{H}_A^{(\ell-1)} - \boldsymbol{P}^{(\ell)}\boldsymbol{W}_B^{(\ell)}\boldsymbol{H}_B^{(\ell-1)}\right\|_F^2\right.$$
$$\left. + \left\langle \boldsymbol{W}_A^{(\ell)}\boldsymbol{H}_A^{(\ell-1)} - \boldsymbol{W}_A^{(\ell)}\boldsymbol{P}^{(\ell-1)}\boldsymbol{H}_B^{(\ell-1)}, \boldsymbol{P}^{(\ell)}\boldsymbol{W}_B^{(\ell)}\boldsymbol{P}^{(\ell-1)^\top}\boldsymbol{H}_A^{(\ell-1)} - \boldsymbol{P}^{(\ell)}\boldsymbol{W}_B^{(\ell)}\boldsymbol{H}_B^{(\ell-1)}\right\rangle_F\right).$$

Consider its first term, i.e.,

$$\underset{\pi=\{\boldsymbol{P}^{(\ell)}\}}{\arg\min} \sum_{\ell=1}^{L}\left\|\boldsymbol{W}_A^{(\ell)}\boldsymbol{H}_A^{(\ell-1)} - \boldsymbol{W}_A^{(\ell)}\boldsymbol{P}^{(\ell-1)}\boldsymbol{H}_B^{(\ell-1)}\right\|_F^2$$

$$\Longleftrightarrow \underset{\pi=\{\boldsymbol{P}^{(\ell)}\}}{\arg\min} \sum_{\ell=1}^{L} \mathrm{tr}\left(\left(\boldsymbol{H}_A^{(\ell-1)^\top}\boldsymbol{W}_A^{(\ell)^\top} - \boldsymbol{H}_B^{(\ell-1)^\top}\boldsymbol{P}^{(\ell-1)^\top}\boldsymbol{W}_A^{(\ell)^\top}\right)\left(\boldsymbol{W}_A^{(\ell)}\boldsymbol{H}_A^{(\ell-1)} - \boldsymbol{W}_A^{(\ell)}\boldsymbol{P}^{(\ell-1)}\boldsymbol{H}_B^{(\ell-1)}\right)\right)$$

$$\Longleftrightarrow \underset{\pi=\{\boldsymbol{P}^{(\ell)}\}}{\arg\min} \sum_{\ell=1}^{L} \mathrm{tr}\left(\boldsymbol{H}_A^{(\ell-1)^\top}\boldsymbol{W}_A^{(\ell)^\top}\boldsymbol{W}_A^{(\ell)}\boldsymbol{H}_A^{(\ell-1)} - \boldsymbol{H}_B^{(\ell-1)^\top}\boldsymbol{P}^{(\ell-1)^\top}\boldsymbol{W}_A^{(\ell)^\top}\boldsymbol{W}_A^{(\ell)}\boldsymbol{H}_A^{(\ell-1)}\right.$$
$$\left. - \boldsymbol{H}_A^{(\ell-1)^\top}\boldsymbol{W}_A^{(\ell)^\top}\boldsymbol{W}_A^{(\ell)}\boldsymbol{P}^{(\ell-1)}\boldsymbol{H}_B^{(\ell-1)} + \boldsymbol{H}_B^{(\ell-1)^\top}\boldsymbol{P}^{(\ell-1)^\top}\boldsymbol{W}_A^{(\ell)^\top}\boldsymbol{W}_A^{(\ell)}\boldsymbol{P}^{(\ell-1)}\boldsymbol{H}_B^{(\ell-1)}\right)$$

$$\Longleftrightarrow \underset{\pi=\{\boldsymbol{P}^{(\ell)}\}}{\arg\min} \sum_{\ell=1}^{L} \mathrm{tr}\left(-2\boldsymbol{H}_B^{(\ell-1)^\top}\boldsymbol{P}^{(\ell-1)^\top}\boldsymbol{W}_A^{(\ell)^\top}\boldsymbol{W}_A^{(\ell)}\boldsymbol{H}_A^{(\ell-1)} + \boldsymbol{H}_B^{(\ell-1)^\top}\boldsymbol{P}^{(\ell-1)^\top}\boldsymbol{W}_A^{(\ell)^\top}\boldsymbol{W}_A^{(\ell)}\boldsymbol{P}^{(\ell-1)}\boldsymbol{H}_B^{(\ell-1)}\right)$$

$$\Longleftrightarrow \underset{\pi=\{\boldsymbol{P}^{(\ell)}\}}{\arg\min} \sum_{\ell=1}^{L} \mathrm{tr}\left(-2\boldsymbol{P}^{(\ell-1)^\top}\boldsymbol{W}_A^{(\ell)^\top}\boldsymbol{W}_A^{(\ell)}\boldsymbol{H}_A^{(\ell-1)}\boldsymbol{H}_B^{(\ell-1)^\top} + \boldsymbol{P}^{(\ell-1)^\top}\boldsymbol{W}_A^{(\ell)^\top}\boldsymbol{W}_A^{(\ell)}\boldsymbol{P}^{(\ell-1)}\boldsymbol{H}_B^{(\ell-1)}\boldsymbol{H}_B^{(\ell-1)^\top}\right)$$

$$\Longleftrightarrow \underset{\pi=\{\boldsymbol{P}^{(\ell)}\}}{\arg\min} \sum_{\ell=1}^{L}\left(\mathrm{tr}\left(\boldsymbol{P}^{(\ell-1)^\top}\boldsymbol{W}_A^{(\ell)^\top}\boldsymbol{W}_A^{(\ell)}\boldsymbol{P}^{(\ell-1)}\boldsymbol{H}_B^{(\ell-1)}\boldsymbol{H}_B^{(\ell-1)^\top}\right) - 2\,\mathrm{tr}\left(\boldsymbol{P}^{(\ell-1)^\top}\boldsymbol{W}_A^{(\ell)^\top}\boldsymbol{W}_A^{(\ell)}\boldsymbol{H}_A^{(\ell-1)}\boldsymbol{H}_B^{(\ell-1)^\top}\right)\right)$$

where $\mathrm{tr}\left(\boldsymbol{P}^{(\ell-1)^\top}\boldsymbol{W}_A^{(\ell)^\top}\boldsymbol{W}_A^{(\ell)}\boldsymbol{P}^{(\ell-1)}\boldsymbol{H}_B^{(\ell-1)}\boldsymbol{H}_B^{(\ell-1)^\top}\right) - 2\mathrm{tr}\left(\boldsymbol{P}^{(\ell-1)^\top}\boldsymbol{W}_A^{(\ell)^\top}\boldsymbol{W}_A^{(\ell)}\boldsymbol{H}_A^{(\ell-1)}\boldsymbol{H}_B^{(\ell-1)^\top}\right)$
is in the form of Koopmans-Beckmann's QAP [14] for each $\boldsymbol{P}^{(\ell-1)}$ and known as NP-hard. Thus, solving $\underset{\pi=\{\boldsymbol{P}^{(\ell)}\}}{\arg\min} \sum_{\ell=1}^{L}\left\|\boldsymbol{W}_A^{(\ell)}\boldsymbol{H}_A^{(\ell-1)} - \boldsymbol{W}_A^{(\ell)}\boldsymbol{P}^{(\ell-1)}\boldsymbol{H}_B^{(\ell-1)}\right\|_F^2$ is to solve $L - 1$ QAPs in parallel.

Similarly, consider the second term, i.e,

$$\arg\min_{\pi=\{\boldsymbol{P}^{(\ell)}\}} \sum_{\ell=1}^{L} \left\| \boldsymbol{P}^{(\ell)} \boldsymbol{W}_B^{(\ell)} \boldsymbol{P}^{(\ell-1)^\top} \boldsymbol{H}_A^{(\ell-1)} - \boldsymbol{P}^{(\ell)} \boldsymbol{W}_B^{(\ell)} \boldsymbol{H}_B^{(\ell-1)} \right\|_F^2$$

$$\iff \arg\min_{\pi=\{\boldsymbol{P}^{(\ell)}\}} \sum_{\ell=1}^{L} \left\| \boldsymbol{W}_B^{(\ell)} \boldsymbol{P}^{(\ell-1)^\top} \boldsymbol{H}_A^{(\ell-1)} - \boldsymbol{W}_B^{(\ell)} \boldsymbol{H}_B^{(\ell-1)} \right\|_F^2$$

$$\iff \arg\min_{\pi=\{\boldsymbol{P}^{(\ell)}\}} \sum_{\ell=1}^{L} \mathrm{tr}\left( \left( \boldsymbol{H}_A^{(\ell-1)^\top} \boldsymbol{P}^{(\ell-1)} \boldsymbol{W}_B^{(\ell)^\top} - \boldsymbol{H}_B^{(\ell-1)^\top} \boldsymbol{W}_B^{(\ell)^\top} \right) \left( \boldsymbol{W}_B^{(\ell)} \boldsymbol{P}^{(\ell-1)^\top} \boldsymbol{H}_A^{(\ell-1)} - \boldsymbol{W}_B^{(\ell)} \boldsymbol{H}_B^{(\ell-1)} \right) \right)$$

$$\iff \arg\min_{\pi=\{\boldsymbol{P}^{(\ell)}\}} \sum_{\ell=1}^{L} \mathrm{tr}\left( \boldsymbol{H}_A^{(\ell-1)^\top} \boldsymbol{P}^{(\ell-1)} \boldsymbol{W}_B^{(\ell)^\top} \boldsymbol{W}_B^{(\ell)} \boldsymbol{P}^{(\ell-1)^\top} \boldsymbol{H}_A^{(\ell-1)} - \boldsymbol{H}_A^{(\ell-1)^\top} \boldsymbol{P}^{(\ell-1)} \boldsymbol{W}_B^{(\ell)^\top} \boldsymbol{W}_B^{(\ell)} \boldsymbol{H}_B^{(\ell-1)} \right.$$

$$\left. - \boldsymbol{H}_B^{(\ell-1)^\top} \boldsymbol{W}_B^{(\ell)^\top} \boldsymbol{W}_B^{(\ell)} \boldsymbol{P}^{(\ell-1)^\top} \boldsymbol{H}_A^{(\ell-1)} + \boldsymbol{H}_B^{(\ell-1)^\top} \boldsymbol{W}_B^{(\ell)^\top} \boldsymbol{W}_B^{(\ell)} \boldsymbol{H}_B^{(\ell-1)} \right)$$

$$\iff \arg\min_{\pi=\{\boldsymbol{P}^{(\ell)}\}} \sum_{\ell=1}^{L} \mathrm{tr}\left( \boldsymbol{H}_A^{(\ell-1)^\top} \boldsymbol{P}^{(\ell-1)} \boldsymbol{W}_B^{(\ell)^\top} \boldsymbol{W}_B^{(\ell)} \boldsymbol{P}^{(\ell-1)^\top} \boldsymbol{H}_A^{(\ell-1)} - 2\boldsymbol{H}_A^{(\ell-1)^\top} \boldsymbol{P}^{(\ell-1)} \boldsymbol{W}_B^{(\ell)^\top} \boldsymbol{W}_B^{(\ell)} \boldsymbol{H}_B^{(\ell-1)} \right)$$

$$\iff \arg\min_{\pi=\{\boldsymbol{P}^{(\ell)}\}} \sum_{\ell=1}^{L} \mathrm{tr}\left( \boldsymbol{P}^{(\ell-1)} \boldsymbol{W}_B^{(\ell)^\top} \boldsymbol{W}_B^{(\ell)} \boldsymbol{P}^{(\ell-1)^\top} \boldsymbol{H}_A^{(\ell-1)} \boldsymbol{H}_A^{(\ell-1)^\top} - 2\boldsymbol{P}^{(\ell-1)} \boldsymbol{W}_B^{(\ell)^\top} \boldsymbol{W}_B^{(\ell)} \boldsymbol{H}_B^{(\ell-1)} \boldsymbol{H}_A^{(\ell-1)^\top} \right)$$

$$\iff \arg\min_{\pi=\{\boldsymbol{P}^{(\ell)}\}} \sum_{\ell=1}^{L} \left( \mathrm{tr}\left( \boldsymbol{P}^{(\ell-1)} \boldsymbol{W}_B^{(\ell)^\top} \boldsymbol{W}_B^{(\ell)} \boldsymbol{P}^{(\ell-1)^\top} \boldsymbol{H}_A^{(\ell-1)} \boldsymbol{H}_A^{(\ell-1)^\top} \right) - 2\,\mathrm{tr}\left( \boldsymbol{P}^{(\ell-1)} \boldsymbol{W}_B^{(\ell)^\top} \boldsymbol{W}_B^{(\ell)} \boldsymbol{H}_B^{(\ell-1)} \boldsymbol{H}_A^{(\ell-1)^\top} \right) \right),$$

which also gives rise to Koopmans-Beckmann's QAPs.

For the last term, i.e,

$$\arg\min_{\pi=\{\boldsymbol{P}^{(\ell)}\}} \sum_{\ell=1}^{L} \left\langle \boldsymbol{W}_A^{(\ell)} \boldsymbol{H}_A^{(\ell-1)} - \boldsymbol{W}_A^{(\ell)} \boldsymbol{P}^{(\ell-1)} \boldsymbol{H}_B^{(\ell-1)}, \boldsymbol{P}^{(\ell)} \boldsymbol{W}_B^{(\ell)} \boldsymbol{P}^{(\ell-1)^\top} \boldsymbol{H}_A^{(\ell-1)} - \boldsymbol{P}^{(\ell)} \boldsymbol{W}_B^{(\ell)} \boldsymbol{H}_B^{(\ell-1)} \right\rangle_F$$

$$\iff \arg\min_{\pi=\{\boldsymbol{P}^{(\ell)}\}} \sum_{\ell=1}^{L} \left\langle \left( \boldsymbol{W}_A^{(\ell)} \boldsymbol{H}_A^{(\ell-1)} - \boldsymbol{W}_A^{(\ell)} \boldsymbol{P}^{(\ell-1)} \boldsymbol{H}_B^{(\ell-1)} \right) \left( \boldsymbol{W}_B^{(\ell)} \boldsymbol{P}^{(\ell-1)^\top} \boldsymbol{H}_A^{(\ell-1)} - \boldsymbol{W}_B^{(\ell)} \boldsymbol{H}_B^{(\ell-1)} \right)^\top, \boldsymbol{P}^{(\ell)} \right\rangle_F,$$

which entails solving bi-level matching problems.

Therefore, the objective can be rewritten as the summation of QAPs and bi-level matching problems and cannot be further simplified, which is NP-hard.

# B More Experimental Details and Results

## B.1 Detailed Experimental Settings

In this section, we introduce the detailed experimental setup. Before delving into details, recall that unless otherwise specified, in this paper we consider models trained on a training set, and then all the investigations are evaluated on a test set.

### B.1.1 Spawning Method

**Multi-Layer Perceptrons on the MNIST Dataset.** In accordance with the settings outlined by Ainsworth et al. [1], we train multi-layer perceptron networks with three hidden layers, each consisting of 512 units, on the MNIST dataset. We adopt the ReLU activation between layers. Optimization is done with the Adam algorithm and a learning rate of $1.2 \times 10^{-4}$. The batch size is set to 60 and the total number of training epochs is 30. To find the modes that satisfy LMC, we start spawning from a common initialization $\boldsymbol{\theta}^{(0)}$.

**VGG-16 and ResNet-20 on the CIFAR-10 Dataset.** In accordance with the settings outlined by Frankle et al. [9], we train the VGG-16 architecture [29] and the ResNet-20 architecture [12] on the CIFAR-10 dataset. Data augmentation techniques include random horizontal flips and random $32 \times 32$ pixel crops. Optimization is done using SGD with momentum (momentum set to 0.9). A weight decay of $1 \times 10^{-4}$ is applied. The learning rate is initialized at 0.1 and is dropped by 10 times at 80 and 120 epochs. The total number of epochs is 160. To find the modes that satisfy LMC, we start spawning after training 5 epochs for both VGG-16 and ResNet-20.

**ResNet-50 on the Tiny-ImageNet Dataset.** In accordance with the settings outlined by Frankle et al. [9], we train the ResNet-50 architecture [12] on the Tiny-ImageNet dataset. Data augmentation techniques include random horizontal flips and random $32 \times 32$ pixel crops. Optimization is done using SGD with momentum (momentum set to 0.9). A weight decay of $1 \times 10^{-4}$ is applied. The learning rate is set to 0.4 and warmed up for 5 epochs and then is dropped by 10 times at 30, 60 and 80 epochs. The total number of epochs is 90. To find the modes that satisfy LMC, we start spawning after training 14 epochs.

### B.1.2 Permutation Method

For the permutation method, we follow the experimental settings of Ainsworth et al. [1] strictly, which are described below.

**Multi-Layer Perceptrons on MNIST and CIFAR-10.** Similar to the spawning method, we use multi-layer perceptron (MLP) networks with three hidden layers, each consisting of 512 units. For MNIST, optimization is performed using Adam with a learning rate of $1 \times 10^{-3}$. For CIFAR-10, optimization is performed using SGD with a learning rate of 0.1. Both activation matching and weight matching are used to identify modes that satisfy LMC.

**ResNet-20 on CIFAR-10.** To achieve LMC, we modify the ResNet-20 architecture by incorporating LayerNorms in place of BatchNorms. Furthermore, we increase the width of ResNet-20 by a factor of 32. Data augmentation techniques include random horizontal flips, random $32 \times 32$ pixel crops, random resizes of the image between $0.8\times$ and $1.2\times$, and random rotations between $\pm 30°$. The optimization process involves using SGD with momentum (set to 0.9). A weight decay regularization term of $5 \times 10^{-4}$ is applied. A single cosine decay schedule with a linear warm-up is applied, where the learning rate is initialized to $1 \times 10^{-6}$ and gradually increased to 0.1 over the course of an epoch, and then a single cosine decay schedule is applied for the remaining training. Only weight matching is used to identify modes that satisfy LMC.

Unlike the spawning method, VGG models are not used in the permutation method due to their inability to achieve LMC. Additionally, Ainsworth et al. [1] open-sourced their source code and pre-trained checkpoints. Therefore, we directly use the pre-trained checkpoints provided by Ainsworth et al. [1].

### B.2 Verification of LLFC Co-Occuring with LMC

In this section, we provide extensive experimental results to verify that LLFC consistently co-occurs with LMC, and conduct a new experiment to demonstrate that the constant $c$ is close to 1 in most cases. Both the spawning method and the permutation method are utilized to obtain linearly connected modes $\boldsymbol{\theta}_A$ and $\boldsymbol{\theta}_B$. As shown in Figures 8 to 13 and 15, we include experimental results for MLP on the MNIST dataset (spawning method, activation matching, and weight matching), MLP on the CIFAR-10 dataset (both activation matching and weight matching), VGG-16 on the CIFAR-10 dataset (spawning method), ResNet-20 on the CIFAR-10 dataset (spawning method and weight matching) and ResNet-50 on the Tiny-ImageNet dataset (spawning method). In particular, in Figure 14, we include experimental results of Straight-Trough Estimator (STE) [1]. STE method tries to learn a permutation with STE that could minimize the loss barrier between one mode and the other permuted mode.

To verify the LLFC property on each data point $\boldsymbol{x}_i$ in the test set $\mathcal{D}$, we measure $\mathrm{cosine}_\alpha(\boldsymbol{x}_i) = \cos[f^{(\ell)}(\alpha \boldsymbol{\theta}_A + (1-\alpha)\boldsymbol{\theta}_B; \boldsymbol{x}_i), \alpha f^{(\ell)}(\boldsymbol{\theta}_A; \boldsymbol{x}_i) + (1-\alpha)f^{(\ell)}(\boldsymbol{\theta}_B; \boldsymbol{x}_i)]$. We compare this to the baseline cosine similarity $\mathrm{cosine}_{A,B}(\boldsymbol{x}_i) = \cos[f^{(\ell)}(\boldsymbol{\theta}_A; \boldsymbol{x}_i), f^{(\ell)}(\boldsymbol{\theta}_B; \boldsymbol{x}_i)]$. In Figures 8 to 15, we

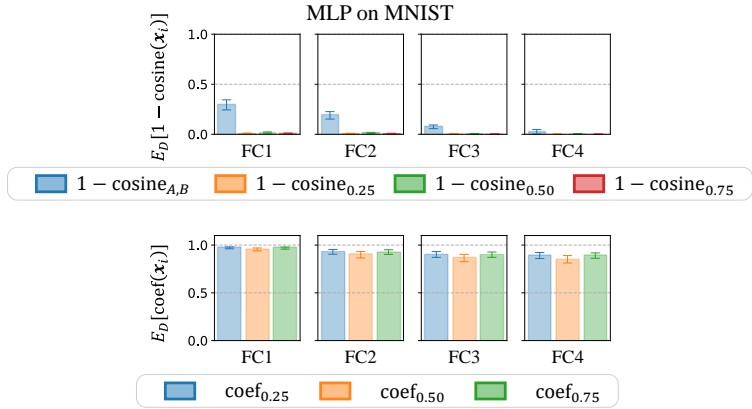

Figure 8: Comparison between $\mathbb{E}_{\mathcal{D}}[1 - \text{cosine}_\alpha(\boldsymbol{x}_i)]$ and $\mathbb{E}_{\mathcal{D}}[1 - \text{cosine}_{A,B}(\boldsymbol{x}_i)]$ and demonstration of $\mathbb{E}_{\mathcal{D}}[1 - \text{coef}_\alpha(\boldsymbol{x}_i)]$. The spawning method is used to obtain two linearly connected modes $\boldsymbol{\theta}_A$ and $\boldsymbol{\theta}_B$. Results are presented for different layers of MLP on MNIST dataset, with $\alpha \in \{0.25, 0.5, 0.75\}$. Standard deviations across the dataset are reported by error bars.

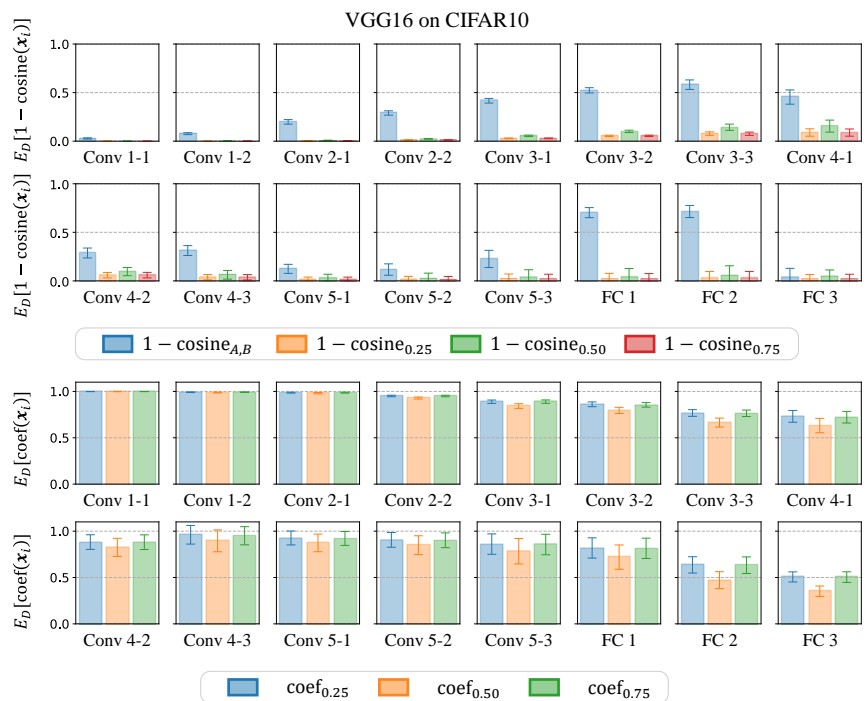

Figure 9: Comparison between $\mathbb{E}_{\mathcal{D}}[1 - \text{cosine}_\alpha(\boldsymbol{x}_i)]$ and $\mathbb{E}_{\mathcal{D}}[1 - \text{cosine}_{A,B}(\boldsymbol{x}_i)]$ and demonstration of $\mathbb{E}_{\mathcal{D}}[1 - \text{coef}_\alpha(\boldsymbol{x}_i)]$. The spawning method is used to obtain two linearly connected modes $\boldsymbol{\theta}_A$ and $\boldsymbol{\theta}_B$. Results are presented for different layers of VGG-16 on the CIFAR-10 dataset, with $\alpha \in \{0.25, 0.5, 0.75\}$. Standard deviations across the dataset are reported by error bars.

conclude that the values of $\mathbb{E}_{\mathcal{D}}[1 - \text{cosine}_\alpha(\boldsymbol{x}_i)]$ are close to 0 compared with $\mathbb{E}_{\mathcal{D}}[1 - \text{cosine}_{A,B}(\boldsymbol{x}_i)]$, and thus verify our claim.

To show that the constant $c$ is close to 1 in most cases, for each data point $\boldsymbol{x}_i$ in the test set $\mathcal{D}$, we measure $\text{coef}_\alpha(\boldsymbol{x}_i) = \|f^{(\ell)}(\alpha\boldsymbol{\theta}_A + (1 - \alpha)\boldsymbol{\theta}_B; \boldsymbol{x}_i)\|\text{cosine}_\alpha(\boldsymbol{x}_i)/\|\alpha f^{(\ell)}(\boldsymbol{\theta}_A; \boldsymbol{x}_i) + (1 - \alpha)f^{(\ell)}(\boldsymbol{\theta}_B; \boldsymbol{x}_i)\|]\|$, where $\|f^{(\ell)}(\alpha\boldsymbol{\theta}_A + (1 - \alpha)\boldsymbol{\theta}_B; \boldsymbol{x}_i)\|\text{cosine}_\alpha(\boldsymbol{x}_i)$ denotes the length of $f^{(\ell)}(\alpha\boldsymbol{\theta}_A + (1 - \alpha)\boldsymbol{\theta}_B; \boldsymbol{x}_i)$ projected on $\alpha f^{(\ell)}(\boldsymbol{\theta}_A; \boldsymbol{x}_i) + (1 - \alpha)f^{(\ell)}(\boldsymbol{\theta}_B; \boldsymbol{x}_i)$. In Figures 8 to 15, we conclude that the values of $\mathbb{E}_{\mathcal{D}}[\text{coef}_\alpha(\boldsymbol{x}_i)]$ are close to 1 in most cases, and thus verify our claim.

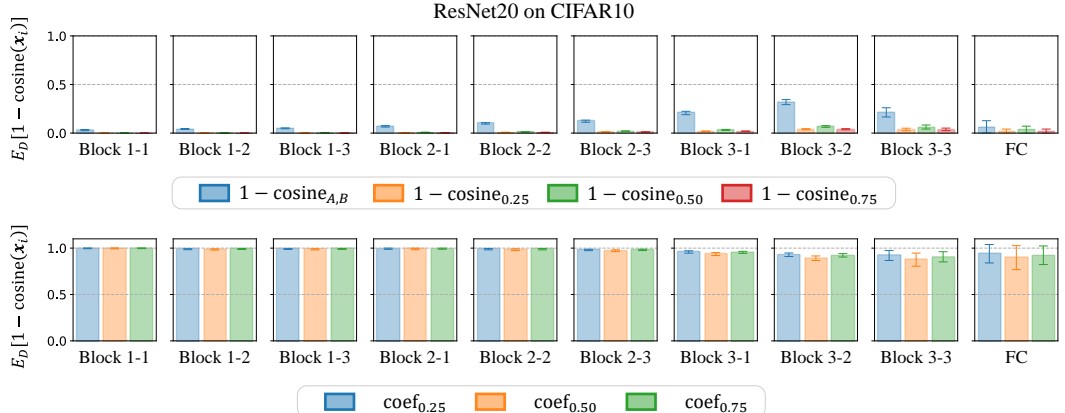

Figure 10: Comparison between $\mathbb{E}_{\mathcal{D}}[1 - \text{cosine}_\alpha(\boldsymbol{x}_i)]$ and $\mathbb{E}_{\mathcal{D}}[1 - \text{cosine}_{A,B}(\boldsymbol{x}_i)]$ and demonstration of $\mathbb{E}_{\mathcal{D}}[1 - \text{coef}_\alpha(\boldsymbol{x}_i)]$. The spawning method is used to obtain two linearly connected modes $\boldsymbol{\theta}_A$ and $\boldsymbol{\theta}_B$. Results are presented for different layers of ResNet-20 on the CIFAR-10 dataset, with $\alpha \in \{0.25, 0.5, 0.75\}$. Standard deviations across the dataset are reported by error bars.

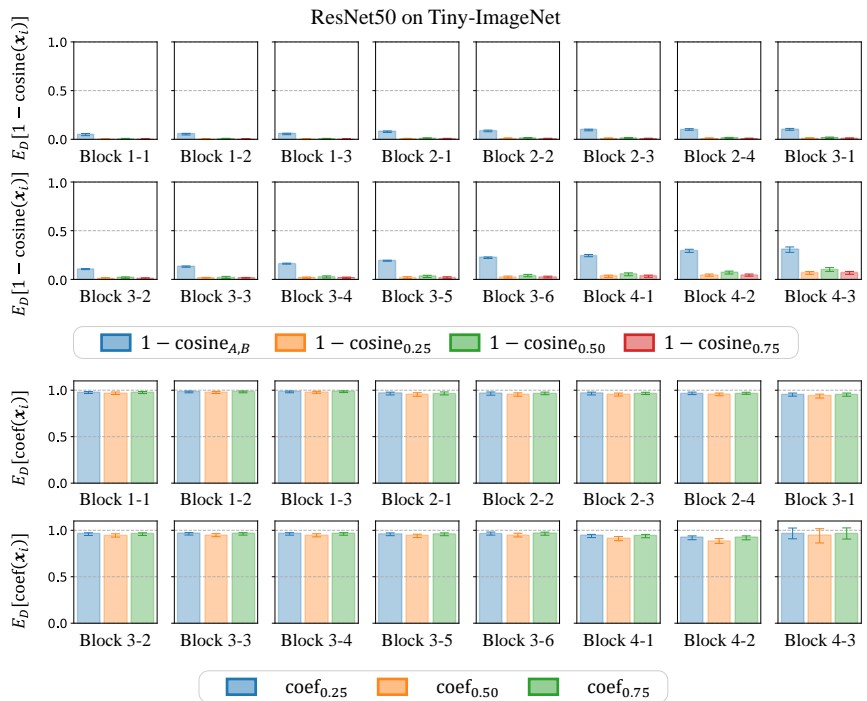

Figure 11: Comparison between $\mathbb{E}_{\mathcal{D}}[1 - \text{cosine}_\alpha(\boldsymbol{x}_i)]$ and $\mathbb{E}_{\mathcal{D}}[1 - \text{cosine}_{A,B}(\boldsymbol{x}_i)]$ and demonstration of $\mathbb{E}_{\mathcal{D}}[1 - \text{coef}_\alpha(\boldsymbol{x}_i)]$. The spawning method is used to obtain two linearly connected modes $\boldsymbol{\theta}_A$ and $\boldsymbol{\theta}_B$. Results are presented for different layers of ResNet-50 on the Tiny-ImageNet dataset, with $\alpha \in \{0.25, 0.5, 0.75\}$. Standard deviations across the dataset are reported by error bars.

## B.3 Verification of Commutativity

In this section, we provide more experimental results on various datasets and model architectures to verify the commutativity property for modes that satisfy LLFC. As shown in Figures 16 to 18, we include more experiments results for VGG-16 on the CIFAR-10 dataset (spawning method), MLP on the MNIST dataset (activation matching) and MLP on the CIFAR-10 dataset (both activation matching and weight matching).

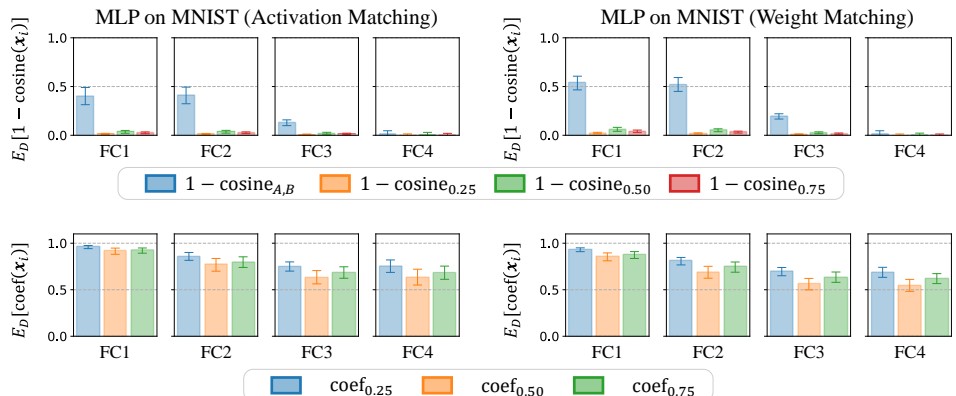

Figure 12: Comparison between $\mathbb{E}_{\mathcal{D}}[1-\text{cosine}_\alpha(\boldsymbol{x}_i)]$ and $\mathbb{E}_{\mathcal{D}}[1-\text{cosine}_{A,B}(\boldsymbol{x}_i)]$ and demonstration of $\mathbb{E}_{\mathcal{D}}[1-\text{coef}_\alpha(\boldsymbol{x}_i)]$. The activation matching and the weight matching are used to obtain two linearly connected modes $\boldsymbol{\theta}_A$ and $\boldsymbol{\theta}_B$. Results are presented for different layers of MLP on the MNIST dataset, with $\alpha \in \{0.25, 0.5, 0.75\}$. Standard deviations across the dataset are reported by error bars.

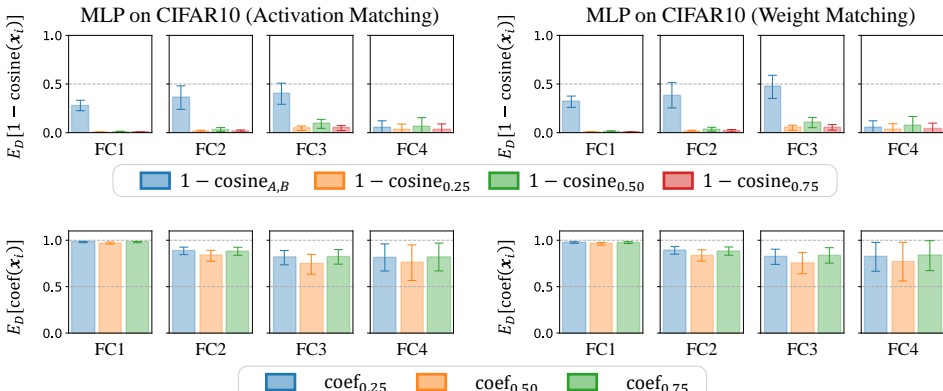

Figure 13: Comparison between $\mathbb{E}_{\mathcal{D}}[1-\text{cosine}_\alpha(\boldsymbol{x}_i)]$ and $\mathbb{E}_{\mathcal{D}}[1-\text{cosine}_{A,B}(\boldsymbol{x}_i)]$ and demonstration of $\mathbb{E}_{\mathcal{D}}[1-\text{coef}_\alpha(\boldsymbol{x}_i)]$. The activation matching and the weight matching are used to obtain two linearly connected modes $\boldsymbol{\theta}_A$ and $\boldsymbol{\theta}_B$. Results are presented for different layers of MLP on the CIFAR-10 dataset, with $\alpha \in \{0.25, 0.5, 0.75\}$. Standard deviations across the dataset are reported by error bars.

To verify the commutativity generally holds for modes that satisfy LLFC, for test set $\mathcal{D}$, we compute $\text{Dist}_{com} = \text{dist}\left(\text{vec}(\boldsymbol{W}_A^{(\ell)}\boldsymbol{H}_A^{(\ell-1)} + \boldsymbol{W}_B^{(\ell)}\boldsymbol{H}_B^{(\ell-1)}), \text{vec}(\boldsymbol{W}_A^{(\ell)}\boldsymbol{H}_B^{(\ell-1)} + \boldsymbol{W}_B^{(\ell)}\boldsymbol{H}_A^{(\ell-1)})\right)$[8]. Furthermore, we compare $\text{Dist}_{com}$ with $\text{Dist}_W = \text{dist}\left(\text{vec}(\boldsymbol{W}_A^{(\ell)}), \text{vec}(\boldsymbol{W}_B^{(\ell)})\right)$ and $\text{Dist}_H = \text{dist}\left(\text{vec}(\boldsymbol{H}_A^{(\ell-1)}), \text{vec}(\boldsymbol{H}_B^{(\ell-1)})\right)$, respectively. In Figures 16 to 18, $\text{Dist}_{com}$ is negligible compared with $\text{Dist}_W$ and $\text{Dist}_H$, confirming the commutativity condition.

Furthermore, we add baselines of models that are not linearly connected to further validate the commutativity condition. In Figure 19, we include experimental results for ResNet-20 on CIFAR-10 dataset (both spawning and weight matching method). Specifically, we measure $\text{Dist}_{com,LMC}$ of two linearly connected modes and $\text{Dist}_{com,not\ LMC}$ of two independently trained modes. In Figure 19, the values of $\text{Dist}_{com,LMC}$ are negligible compared with $\text{Dist}_{com,not\ LMC}$, which confirms the commutativity condition.

---

[8]We also conduct experiments on CNNs. For a Conv layer, the forward propagation will be denoted as $\boldsymbol{W}\boldsymbol{H}$ similar to a linear layer.

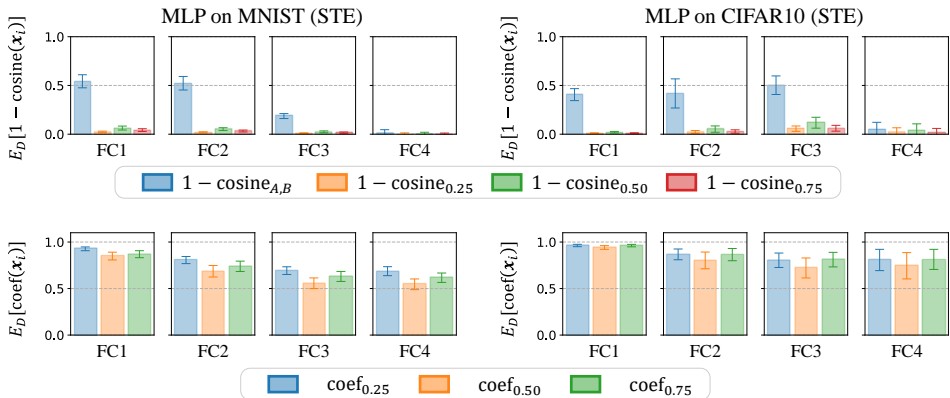

Figure 14: Comparison between $\mathbb{E}_{\mathcal{D}}[1-\text{cosine}_\alpha(\boldsymbol{x}_i)]$ and $\mathbb{E}_{\mathcal{D}}[1-\text{cosine}_{A,B}(\boldsymbol{x}_i)]$ and demonstration of $\mathbb{E}_{\mathcal{D}}[1-\text{coef}_\alpha(\boldsymbol{x}_i)]$. The Straight-Through Estimator (STE) [1] are used to obtain two linearly connected modes $\boldsymbol{\theta}_A$ and $\boldsymbol{\theta}_B$. Results are presented for different layers of MLP on both MNIST and CIFAR-10 dataset, with $\alpha \in \{0.25, 0.5, 0.75\}$. Standard deviations across the dataset are reported by error bars.

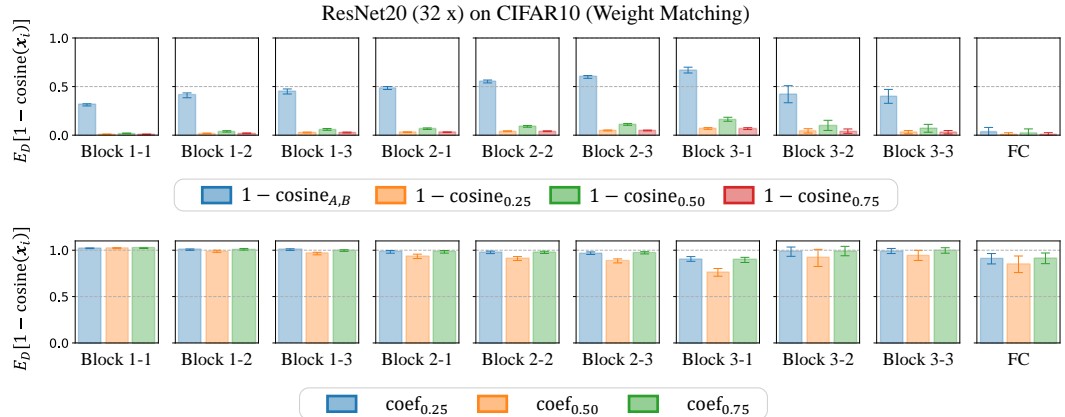

Figure 15: Comparison between $\mathbb{E}_{\mathcal{D}}[1-\text{cosine}_\alpha(\boldsymbol{x}_i)]$ and $\mathbb{E}_{\mathcal{D}}[1-\text{cosine}_{A,B}(\boldsymbol{x}_i)]$ and demonstration of $\mathbb{E}_{\mathcal{D}}[1-\text{coef}_\alpha(\boldsymbol{x}_i)]$. The weight matching is used to obtain two linearly connected modes $\boldsymbol{\theta}_A$ and $\boldsymbol{\theta}_B$. Results are presented for different layers of ResNet-20 (32x) on the CIFAR-10 dataset, with $\alpha \in \{0.25, 0.5, 0.75\}$. Standard deviations across the dataset are reported by error bars.

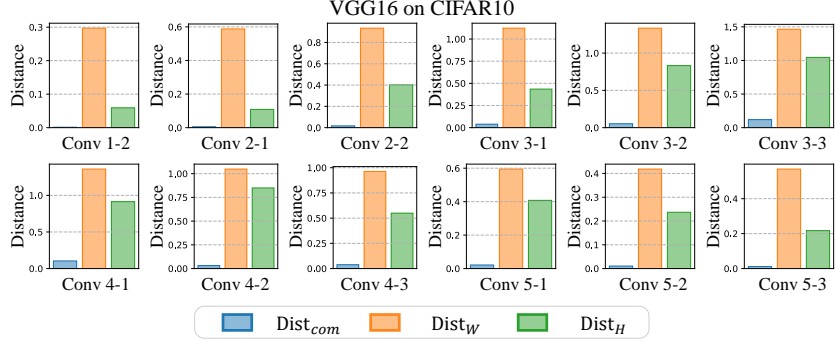

Figure 16: Comparison of $\text{Dist}_{com}$, $\text{Dist}_W$, and $\text{Dist}_H$. The spawning method is used to obtain two modes that satisfy LLFC, $\boldsymbol{\theta}_A$ and $\boldsymbol{\theta}_B$. The results are presented for different layers of VGG-16 on the CIFAR-10 dataset.

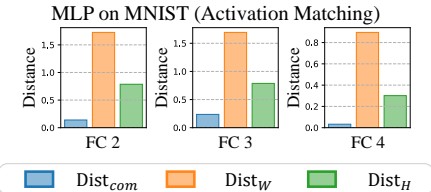

Figure 17: Comparison of $\text{Dist}_{com}$, $\text{Dist}_W$, and $\text{Dist}_H$. The activation matching is used to obtain two modes that satisfy LLFC, $\boldsymbol{\theta}_A$ and $\boldsymbol{\theta}_B$. The results are presented for different layers of MLP on the MNIST dataset.

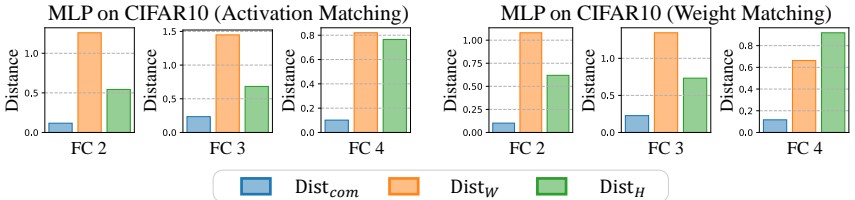

Figure 18: Comparison of $\text{Dist}_{com}$, $\text{Dist}_W$, and $\text{Dist}_H$. Both the activation matching and weight matching are used to obtain two modes that satisfy LLFC, $\boldsymbol{\theta}_A$ and $\boldsymbol{\theta}_B$. The results are presented for different layers of MLP on the CIFAR10 dataset.

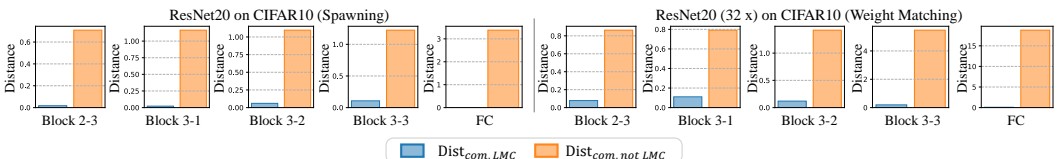

Figure 19: Comparison between $\text{Dist}_{com,LMC}$ and $\text{Dist}_{com,not\ LMC}$. Both the spawning and permutation methods are used to obtain two linearly connected modes.

| Layer $\ell$ | FC 1 | FC 2 | FC 3 |
|---|---|---|---|
| $\text{Err}_{\mathcal{D}(B_{>\ell}\circ A_{\leq\ell})}$ | 2.69 | 2.11 | 1.92 |

Table 1: Error rates (%) of stitched MLP on the MNIST test set. The model stitching is employed in different layers. The spawning method is used to obtain two neural networks that satisfy LLFC, i.e., $A$ and $B$. Error rates (%) of $A$ and $B$ are 1.9 and 1.77, respectively.

Notably, the experiments are not conducted on the first Conv/Linear layer of the model because the commutativity condition is naturally satisfied for the first layer where $\boldsymbol{H}_A^{(0)} = \boldsymbol{H}_B^{(0)} = \boldsymbol{X}$ where $\boldsymbol{X}$ is the input data matrix.

### B.4 Experiments on Model Stitching

Model stitching [19, 3] is commonly employed to analyze neural networks' internal representations. Let $A$ and $B$ represent neural networks with identical architectures. Given a loss function $\mathcal{L}$, model stitching involves finding a stitching layer $s$ (e.g., a linear $1 \times 1$ convolutional layer) such that the minimization of $\mathcal{L}(B_{>\ell} \circ s \circ A_{\leq\ell})$ is achieved. Here, $B_{>\ell}$ denotes the mapping from the activations of the $\ell$-th layer of network $B$ to the final output, $A_{\leq\ell}$ denotes the mapping from the input to the activations of the $\ell$-th layer of network $A$, and $\circ$ represents function composition.

In this section, we explore a stronger form of model stitching. Specifically, given two neural networks $A$ and $B$ that satisfy LLFC, we evaluate the arruacy of $B_{>\ell} \circ A_{\leq\ell}$ over the test set $\mathcal{D}$ without finding a stitching layer, i.e., $\text{Err}_{\mathcal{D}(B_{>\ell}\circ A_{\leq\ell})}$. As shown in Tables 1 to 3, we include experimental results for MLP on the MNIST dataset, VGG-16 on CIFAR-10 the dataset and ResNet-20 on the CIFAR-10 dataset. Only the spawning method is utilized to find modes that satisfy LLFC. The results depicted

| Layer $\ell$ | Conv 1-1 | Conv 1-2 | Conv 2-1 | Conv 2-2 | Conv 3-1 |
|---|---|---|---|---|---|
| $\mathrm{Err}_{\mathcal{D}(B_{>\ell}\circ A_{\leq\ell})}$ | 7.2 | 8.43 | 8.39 | 9.91 | 11.84 |
| Layer $\ell$ | Conv 3-2 | Conv 3-3 | Conv 4-1 | Conv 4-2 | Conv 4-3 |
| $\mathrm{Err}_{\mathcal{D}(B_{>\ell}\circ A_{\leq\ell})}$ | 9.55 | 8.22 | 7.61 | 6.99 | 7.05 |
| Layer $\ell$ | Conv 5-1 | Conv 5-2 | Conv 5-3 | FC 1 | FC 2 |
| $\mathrm{Err}_{\mathcal{D}(B_{>\ell}\circ A_{\leq\ell})}$ | 6.91 | 6.88 | 6.88 | 7.07 | 6.92 |

Table 2: Error rates (%) of stitched VGG-16 on the CIFAR-10 test set. The model stitching is employed in different layers. The spawning method is used to obtain two neural networks that satisfy LLFC, i.e., $A$ and $B$. Error rates (%) of $A$ and $B$ are $6.87$ and $7.1$, respectively.

| Layer $\ell$ | Block 1-1 | Block 1-2 | Block 1-3 | Block 2-1 | Block 2-2 | Block 2-3 | Block 3-1 | Block 3-2 | Block 3-3 |
|---|---|---|---|---|---|---|---|---|---|
| $\mathrm{Err}_{\mathcal{D}(B_{>\ell}\circ A_{\leq\ell})}$ | 10.88 | 10.57 | 13.35 | 10.64 | 10.74 | 10.55 | 12.27 | 11.8 | 8.99 |

Table 3: Error rates (%) of stitched ResNet-20 on the CIFAR-10 test set. The model stitching is employed in different layers. The spawning method is used to obtain two neural networks that satisfy LLFC, i.e., $A$ and $B$. Error rates (%) of $A$ and $B$ are $8.69$ and $8.58$, respectively.

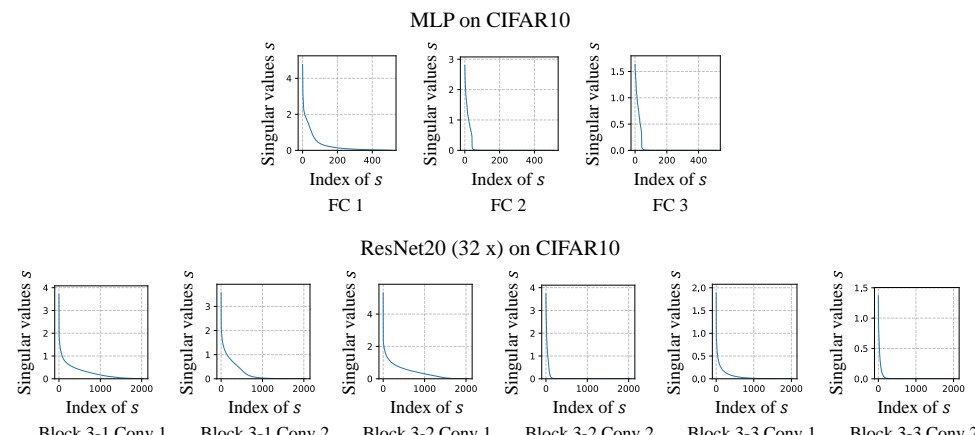

Figure 20: Singular values of weight matrix $\boldsymbol{W}^{(\ell)}$ of $\ell$-th layer of $\boldsymbol{\theta}$ in a descending order. Here, $\boldsymbol{\theta}$ can be used to achieve LMC with weight matching. The results are presented for different layers of various model architectures and datasets.

in Tables 1 to 3 demonstrate that the error rates of the stitched model on the test set closely resemble the error rates of the original models $A$ and $B$, regardless of the dataset or model architecture. This observation suggests that models that satisfy LLFC encode similar information, which can be decoded across different models. Subsequently, the experiments of model stitching provides new insights towards the commutativity property, i.e, $\forall \ell \in [L], \boldsymbol{W}_B^{(\ell)} \boldsymbol{H}_A^{(\ell-1)} \approx \boldsymbol{W}_B^{(\ell)} \boldsymbol{H}_B^{(\ell-1)}$.

## B.5 Discussion on Git Re-basin [1]

In this section, we investigate the ability of permutation methods to achieve LMC. While we have interpreted the activation matching and weight matching methods proposed by Ainsworth et al. [1] as guaranteeing the commutativity property, we have yet to address why permutation methods can ensure the satisfaction of this property. Thus, in order to delve into the capability of permutation methods, we must address the question of why these methods are capable of ensuring the satisfaction of the commutativity property.

Low-rank model weights and activations contribute to ensure the commutativity property. We now consider a stronger form of the commutativity property, where given two modes $\boldsymbol{\theta}_A$ and $\boldsymbol{\theta}_B$ and a

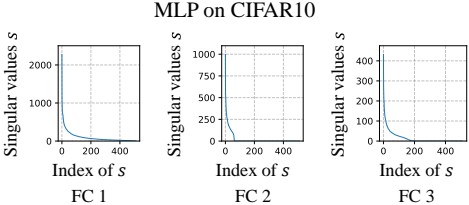

Figure 21: Singular values of post-activations $\boldsymbol{H}^{(\ell)}$ of $\ell$-th layer of $\boldsymbol{\theta}$ over the whole test set $\mathcal{D}$ in a descending order. Here, $\boldsymbol{\theta}$ can be used to achieve LMC with activation matching.The results are presented for different layers of MLP on the CIFAR-10 dataset.

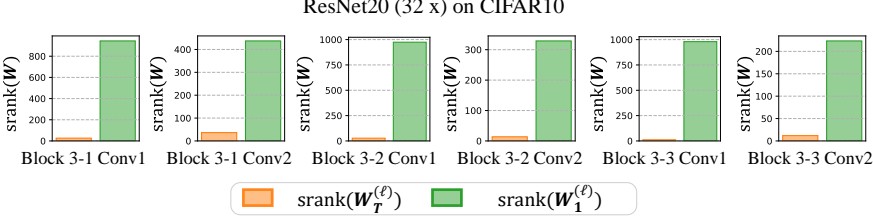

Figure 22: Comparion between the stable rank $\mathrm{srank}(\boldsymbol{W}_T^{(\ell)})$ and $\mathrm{srank}(\boldsymbol{W}_1^{(\ell)})$. Here, $\boldsymbol{W}_T^{(\ell)}$ denotes the weight matrix of the $\ell$-th layer of the model $\boldsymbol{\theta}_T$ in the terminal phase of training. Similarly, $\boldsymbol{W}_1^{(\ell)}$ denotes the weight matrix of the $\ell$-th layer of the model $\boldsymbol{\theta}_1$ in the early stage of training (1 epoch indeed). Also, the stable rank can be calculated as $\mathrm{srank}(\boldsymbol{W}) = \frac{\|\boldsymbol{W}\|_F^2}{\|\boldsymbol{W}\|_2^2}$. The results are presented for different layers of ResNet-20 (32x) on the CIFAR-10 dataset.

dataset $\mathcal{D}$, we have:

$$\forall \ell \in [L], \boldsymbol{W}_A^{(\ell)} \boldsymbol{H}_A^{(\ell-1)} = \boldsymbol{W}_A^{(\ell)} \boldsymbol{H}_B^{(\ell-1)} \wedge \boldsymbol{W}_B^{(\ell)} \boldsymbol{H}_B^{(\ell-1)} = \boldsymbol{W}_B^{(\ell)} \boldsymbol{H}_A^{(\ell-1)}.$$

Thus, to satisfy the commutativity property for a given layer $\ell$, we can employ the permutation method to find a permutation matrix $\boldsymbol{P}^{(\ell-1)}$ such that:

$$\boldsymbol{W}_A^{(\ell)} \left( \boldsymbol{H}_A^{(\ell-1)} - \boldsymbol{P}^{(\ell-1)} \boldsymbol{H}_B^{(\ell-1)} \right) = 0 \wedge \boldsymbol{P}^{(\ell)} \boldsymbol{W}_B^{(\ell)} \left( \boldsymbol{H}_B^{(\ell-1)} - \boldsymbol{P}^{(\ell-1)^\top} \boldsymbol{H}_A^{(\ell-1)} \right) = 0.$$

In a homogeneous linear system $\boldsymbol{W}\boldsymbol{X} = 0$, a low-rank matrix $\boldsymbol{W}$ allows for a larger solution space for $\boldsymbol{X}$. Therefore, if the ranks of $\boldsymbol{W}_A^{(\ell)}$ and $\boldsymbol{W}_B^{(\ell)}$ are low, it becomes easier to find a permutation matrix $\boldsymbol{P}^{(\ell-1)}$ that satisfies the commutativity property. Similarly, if we consider another form of commutativity property:

$$\forall \ell \in [L], \boldsymbol{W}_A^{(\ell)} \boldsymbol{H}_A^{(\ell-1)} = \boldsymbol{W}_B^{(\ell)} \boldsymbol{H}_A^{(\ell-1)} \wedge \boldsymbol{W}_B^{(\ell)} \boldsymbol{H}_B^{(\ell-1)} = \boldsymbol{W}_A^{(\ell)} \boldsymbol{H}_B^{(\ell-1)}.$$

Then, to ensure the commutativity property, we need to find $\boldsymbol{P}^{(\ell-1)}$ and $\boldsymbol{P}^{(\ell)}$ such that

$$\left( \boldsymbol{W}_A^{(\ell)} - \boldsymbol{P}^{(\ell)} \boldsymbol{W}_B^{(\ell)} \boldsymbol{P}^{(\ell-1)^\top} \right) \boldsymbol{H}_A^{(\ell-1)} = 0 \wedge \left( \boldsymbol{P}^{(\ell)} \boldsymbol{W}_B^{(\ell)} \boldsymbol{P}^{(\ell-1)^\top} - \boldsymbol{W}_A^{(\ell)} \right) \boldsymbol{P}^{(\ell-1)} \boldsymbol{H}_B^{(\ell-1)} = 0.$$

Then, if the ranks of $\boldsymbol{H}_A^{(\ell-1)}$ and $\boldsymbol{H}_B^{(\ell-1)}$ are low, it is easier to find the permutation matrices to satisfy the condition. In real scenarios, both model weights (see Figure 20) and activations (see Figure 21) are approximately low-rank, which helps the permutation methods satisfy the commutativity property.

Additionally, Ainsworth et al. [1] mentioned two instances where permutation methods can fail: models with insufficient width and models in the early stages of training. In both cases, the model weights often fail to satisfy the low-rank model weight condition. In the first scenario, when the model lacks sufficient width, meaning that the dimension of the weight matrix approaches the rank of the weight matrix, the low-rank condition may not be met. For example, compared the singular values of ResNet-20 (32x) (see Figure 20) with singular values of ResNet-20 (1x) (see **??**), it is evident that in the wider architecture, the proportion of salient singular values is smaller. In the second scenario, during the initial stages of training, the weight matrices resemble random matrices and may not

exhibit low-rank characteristics. For example, as shown in Figure 22, the stable ranks of weight matrices of the model after convergence are significantly smaller than those of the model in the early stage of training. Consequently, permutation methods may struggle to find suitable permutations that fulfill the commutativity property, resulting in the inability to obtain modes that satisfy LMC.

