# OpenReview forum: "Going Beyond Linear Mode Connectivity: The Layerwise Linear Feature Connectivity"
_NeurIPS.cc/2023/Conference — NeurIPS 2023 poster_

### Official Review · Reviewer_Vyah · 2023-07-04

**Soundness:** 3 good
**Presentation:** 4 excellent
**Contribution:** 3 good
**Rating:** 7
**Confidence:** 3

**Summary:**

There is a continuing effort to understand the complex training dynamics and loss landscape of neural networks, and one of the most interesting discoveries is Linear Mode Connectivity (LMC). LMC is the phenomenon that when two different solutions are linearly interpolated in parameter space, the training loss and test loss remain low enough to be similar to the solution. In this paper, the authors propose Layerwise Linear Feature Connectivity (LLFC), a stronger linear connectivity than LMC. LLFC means that the feature maps of all layers of two differently trained solutions are also linearly connected. Through various experiments, they show that LLFC is satisfied when spawning and permutation methods, which are common methods for presenting LMC characteristics, are used. In addition, they advance the understanding of LMC with a theoretical explanation of the underlying factors that make LLFC appear naturally.

**Strengths:**

- The paper introduced LLFC, an interesting phenomenon that extends LMC, linear connectivity in parameter space, to linear connectivity in feature space.
- The empirical results are comprehensive and contain the necessary contents for the development of the discussion.
- The theoretical analysis provides a convincing explanation for the emergence of LLFC.

**Weaknesses:**

- The authors only ran their experiments on MNIST and CIFAR10, which are relatively easy datasets. It will be important to verify its validity on larger datasets such as ImageNet. Git Re-Basin (Ainsworth at el., 2023) also showed a relatively large loss barrier on ImageNet, so validation on a wider range of datasets is required.
-----
(Ainsworth at el., 2023) [Git Re-Basin: Merging Models modulo Permutation Symmetries](https://openreview.net/forum?id=CQsmMYmlP5T)

**Questions:**

- I'm not sure how low the value of the sparsity measure is in Figure 4. Is a value of ~0.3 considered sparse?
- In the experiments, the values measured with random variables, etc. are used as baseline, but it would be more accurate to compare them to the values for a model that is not linearly connected. The baseline values currently presented seem to exaggerate the results because they are comparing too different components.
- Just a question, do you think the ensemble performance would be improved when ensembling linearly interpolated features with the modes?

**Limitations:**

I see no potential negative societal impact from this paper.

---

> ### Author Rebuttal · Authors · 2023-08-09
>
> **Q1: Ask for additional experiments on larger datasets such as ImageNet. “The authors only ran their experiments on MNIST and CIFAR10, which are *relatively easy* datasets. It will be important to verify its validity on larger datasets such as ImageNet.”**
>
> **A1**: Thank you for great suggestions. To begin with, we want to clarify that the purpose of our main experiments is to verify that LLFC co-occurs with LMC. Because the permutation methods cannot achieve the zero-loss-barrier LMC on ImageNet dataset [1], we can not verify that LLFC co-occurs with LMC in this particular instance. However, we take note that Frankle et al. [8] observed LMC on ResNet-50 trained on ImageNet dataset using the spawning method. Therefore, we follow your suggestion and conduct additional experiments on larger datasets using the spawning method. Given the temporal constraints of the Rebuttal period and the limitations of computational resources, we opted for the Tiny-ImageNet dataset [Cite 1], which is smaller but harder challenge than ImageNet.
>
> Our experiments adhere to the identical training configurations as those outlined in Frankle et al. [8]. We apply spawning method to obtain the two linearly connected modes, $\boldsymbol{\theta_A}$ and $\boldsymbol{\theta}_B$. Subsequently, consistent with the experimental settings of the main paper, we evaluate both ${\rm cosine}\_{\alpha}(\boldsymbol{x}_i)$ and ${\rm cosine}\_{A, B}(\boldsymbol{x}_i)$ for each data point $\boldsymbol{x}_i$ in the test set $\mathcal{D}$. In Figure 1 (global response), the values of $\mathbb{E}\_{\mathcal{D}}[1-{\rm cosine}\_{\alpha}(\boldsymbol{x}_i)]$ consistently approximate zero in contrast to $\mathbb{E}\_{\mathcal{D}}[1-{\rm cosine}\_{A,B}(\boldsymbol{x}_i)]$ across different layers and different values of $\alpha$. This further strengthens our argument, confirming the co-occurrence of LLFC and LMC. For more experiment details, please kindly refer to our global response.
>
> [cite 1] Le and Xuan Yang. Tiny imagenet visual recognition challenge. CS 231N, 7(7):3, 2015
>
> **Q2: Ask for a baseline of sparsity for comparison. “I'm not sure how low the value of the sparsity measure is in Figure 4.”**
>
> **A2**: Thank you for great question. We follow your suggestion and conduct new experiments to add baselines of sparsity for comparison. We choose the pre-activations of random initialized networks as our baseline. We measure the sparsity of the pre-activations of both well-trained networks and random initialized networks, using $S(\boldsymbol{x}) = \frac{\|\boldsymbol{x}\|_1}{n\|\boldsymbol{x}\|\_{\infty}}(\boldsymbol{x} \in \mathbb{R}^n)$, denoted as $S(\tilde{\boldsymbol{h}}\_{i, \text{end}})$ and $S(\tilde{\boldsymbol{h}}\_{i, \text{init}})$ respectively. In Figure 3 (global response), the values of $S(\tilde{\boldsymbol{h}}\_{i, \text{end}})$ are relatively small compared to $S(\tilde{\boldsymbol{h}}\_{i, \text{init}})$, thus providing further supports for our sparsity claim. For more experimental details, please kindly refer to our global response.
>
> **Q3: Ask for more baseline for verifying weak additivity condition. “In the experiments, the values measured with random variables, etc. are used as baseline, but it would be more accurate to compare them to the values for a model that is not linearly connected.”**
>
> **A3**: Thank you for your great suggestion. We follow your suggestion and conduct additional experiments to comparing with models that are not linearly connected. Specifically, we compare $\text{Dist}\_{\sigma}(\tilde{\boldsymbol{h}}\_{i, A}, \tilde{\boldsymbol{h}}\_{i, B})$ with $\text{Dist}\_{\sigma}(\tilde{\boldsymbol{h}}\_{i, C}, \tilde{\boldsymbol{h}}\_{i, D})$ and $\text{Dist}\_{\sigma}(\boldsymbol{r}_1, \boldsymbol{r}_2)$. Here, $\tilde{\boldsymbol{h}}\_{i, A}, \tilde{\boldsymbol{h}}\_{i, B}$ denote the pre-activations of two linearly connected mode $\boldsymbol{\theta}\_A$ and $\boldsymbol{\theta}\_B$, while $\tilde{\boldsymbol{h}}\_{i, C}, \tilde{\boldsymbol{h}}\_{i, D}$ denote the pre-activations of two independently trained mode $\boldsymbol{\theta}_C$ and $\boldsymbol{\theta}_D$.  Meanwhile, $\boldsymbol{r}_1$ and $\boldsymbol{r}_2$ are still independent $d\_{\ell}$-dimensional random vectors sampled from $\mathcal{N}(\boldsymbol{0}, \boldsymbol{I})$. In Figure 3 (global response), the values of $\text{Dist}\_{\sigma}(\tilde{\boldsymbol{h}}\_{i, A}, \tilde{\boldsymbol{h}}\_{i, B})$ are negligible in comparison to $\text{Dist}\_{\sigma}(\tilde{\boldsymbol{h}}\_{i, C}, \tilde{\boldsymbol{h}}\_{i, D})$ and $\text{Dist}\_{\sigma}(\boldsymbol{r}_1, \boldsymbol{r}_2)$, which further validate the weak additivity condition for linearly connected models. For more experiment details, please kindly refer to our global response.
>
> **Q4: Question on ensemble performance. “Just a question, do you think the ensemble performance would be improved when ensembling linearly interpolated features with the modes?”**
>
> **A4**: Thank you for the interesting question. While we do not have a definitive answer to this question, we do wish to highlight several studies that bear relevance. For example, as explored in Section 5.4 of Ainsworth et al. [1], an interesting observation arises when training two models on distinct splits of the dataset. In this context, the performance of a linearly interpolated model fails to match that of an ensemble of two models with twice the number of effective weights. This intriguing issue holds potential for further investigation and we can get insights to enhance the design of more effective ensemble methods.

---

> > ### Comment · Reviewer_Vyah · 2023-08-14
> >
> > Thank you for the clarification and revisions to the result as I suggest. I believe that LLFC is an interesting phenomenon and will be helpful to many researchers.

---

### Official Review · Reviewer_4GwW · 2023-07-04

**Soundness:** 3 good
**Presentation:** 4 excellent
**Contribution:** 2 fair
**Rating:** 5
**Confidence:** 4

**Summary:**

The work identifies layerwise  linear feature connectivity (LLFC) and proves LLFC is sufficient for linear mode connectivity (LMC) between neural networks. Experimental evidence using two methods for finding LMC networks (spawning and permutation) finds that LMC and LLFC co-occur in a variety of models and datasets. Two conditions (weak additivity of ReLU and commutativity) are identified as sufficient for LLFC, and experimental evidence is presented for these conditions occuring in LMC networks.

**Strengths:**

The paper unifies two lines of research into LMC (spawning versus permutation methods) under a common mathematical framework. The notion of commutativity is an interesting derivation and not as obvious as it seems at first glance. The paper is well written and easy to follow, with clear definitions. If established comprehensively, LLFC would be a significantly stronger condition than LMC, and would have far-reaching implications.

**Weaknesses:**

Main issues:
- Current alignment algorithms for permuting LMC networks already directly optimize for LLFC (section 5.3), so the finding that LLFC implies LMC (Lemma 1) is formalizing a well-established phenomenon which is somewhat obvious.
- As a result, LLFC implying LMC is somehow less interesting than LMC implying LLFC, but the latter direction is not explored. Only co-occurence is observed experimentally, making the direction of causation unclear. In particular, if LMC is found to imply LLFC, the reasons are likely to be extremely informative.
- For spawning LMC networks, it is unclear how the early training epochs contribute to enabling LMC. The experiments do not consider the evolution of LMC networks through training time. Furthermore, the connection between figure 6 and the equation in lines 285-286 is unclear (see questions below).

Additionally, the experimental evidence lacks baselines for comparison. In general, a fair comparison should include interpolated non-LMC networks.
- Figure 2 and 3: no cosine similarity for interpolated networks that are not linearly connected. Since interpolating between any vectors (including random ones) increases their cosine similarity considerably, it is not clear that the increase in similarity is due to LLFC and not averaging.
- Figure 4: no sparsity baseline for random networks. Currently it is unclear what is considered "small" in line 232.
- Figure 5: no commutativity distance for non-LMC networks

**Questions:**

- Section 5.2: how  does similarity of $U_A$ and $U_B$ imply commutativity when $W = U \Sigma V$ also depends on $V$, which could differ significantly between two networks? The connection between figure 6 and the equation in lines 285-286 is unclear.
- How much would solving the quadratic assignment problem improve over existing algorithms? Given that activation similarity is closely correlated with weight similarity, it is not obvious whether minimizing the QAP will lead to significant improvements in permutation alignment versus aligning weights and/or alignments, which is equivalent to minimizing one or both sides of equation 9 separately.
- Weak additivity seems to rely heavily on the ReLU function. Does LLFC occur for other activation functions?
- Do networks exist where LMC holds but not LLFC? If so, how can one find such networks and how common are they?

**Limitations:**

Overall the findings could be explored in greater depth and better related to observable phenomena. While LLFC is shown to imply LMC, the reverse is not shown or refuted. For spawned LMC networks, it is not clear whether LLFC is a byproduct of averaging, how LLFC relates to the onset of LMC at an early point in training, or how LLFC can be increased/decreased. For permuted LMC networks, current alignment algorithms (e.g. Ainsworth et al. 2022, Singh & Jaggi 2020) are already explicitly optimizing for LLFC (via weight or activation alignment), and thus the findings have limited explanatory power.

---

> ### Author Rebuttal · Authors · 2023-08-09
>
> **Q1: “Current alignment algorithms for permuting LMC networks already directly optimize for LLFC (section 5.3), so the finding that LLFC implies LMC (Lemma 1) is formalizing a well-established phenomenon which is somewhat obvious.”**
>
> **A1**: We'd like to emphasize that the interpretation that current alignment algorithms optimize for LLFC (Sec 5.3) is a new finding and one of the key contributions of this paper. This provides a deeper theoretical support for the alignment algorithms, which was absent in existing literature. See Reviewer Dt29's comments: "The fact that both the weight-matching and activation-matching losses can be found in the commutativity condition is quite neat, as (to the best of our knowledge) these two methods were proposed as heuristics." We respectfully disagree that any of our findings were well-established, since the notion of LLFC never appeared in any prior work.
>
> We'd also like to clarify the main contributions and logic progression of this paper:
>
> - We formulate LLFC, which is a stronger generalization of LMC (Lemma 1).
> - We empirically find that LMC networks always satisfy LLFC in practice, without exception (Sec 4).
> - We identify two conditions (weak additivity and commutativity) which imply LLFC, and verify them empirically.
>
> Together, our work establishes LLFC as a more fine-grained and fundamental phenomenon than LMC. This novel perspective contributes to the new understanding of LMC and the spawning and permutation methods. Lemma 1 is only to formally verify that LLFC is stronger than LMC, but is not the main technical contribution of this paper.
>
> **Q2: “LLFC implying LMC is somehow less interesting than LMC implying LLFC, but the latter direction is not explored. Only co-occurence is observed experimentally, making the direction of causation unclear…”**
>
> **A2**: We have done a significant amount of experiments to verify "LMC implies LLFC in practice", i.e., all existing LMC networks satisfy LLFC, without exception (Sec 4). This shows that LLFC is a more fundamental phenomenon that may be the underlying cause of LMC in practice, and we make use of this perspective to obtain new insights (Sec 5). For the reverse direction, it is unlikely to establish LMC implies LLFC unconditionally, because LMC is a global property that only concerns the network output and LLFC characterizes the finer details in each layer in the network.
>
> **Q3: “For spawning LMC networks, it is unclear how the early training epochs contribute to enabling LMC. The experiments do not consider the evolution of LMC networks through training time.” and “how LLFC relates to the onset of LMC at an early point in training”**
>
> **A3**: Thank you for the comment. In Sec 5.2, we find that two spawned networks share similar principal directions in model weights, which contributes to the satisfaction of commutativity condition, thus leading to the satisfaction of LLFC and thereby LMC. This indicates that the early training epochs determine the top principal directions of the weights. In light of your suggestion, as shown in Figure 6 (global response), we conduct new experiments to evaluate the relationship between the number of shared early training epochs and the similarity between principal directions of spawned networks. We kindly refer you to the global response for more experimental details.
>
> **Q4: About baselines for comparison. “Figure 2 and 3: no cosine similarity for interpolated networks that are not linearly connected…”, “Figure 4: no sparsity baseline for random networks…” and “Figure 5: no commutativity distance for non-LMC networks…”**
>
> **A4**: Thank you for your suggestions. As shown in Figure 2 to 4 (global response), we conduct new experiments to add baselines for comparison. We kindly refer you to the global response for more experimental details.
>
> **Q5: “Section 5.2: how does similarity of $U_A$ and $U_B$ imply commutativity when $W=U\Sigma V$ also depends on $V$, which could differ significantly between two networks?...”**
>
> **A5**: Thank you for your comment. As shown in Figure 5 (global response), we have verified the similarity between $V_A$ and $V_B$, and had the same observation as $U$. We kindly refer you to the global response for the experimental details. We will add this to the final version.
>
> **Q6: “How much would solving the quadratic assignment problem improve over existing algorithms? …it is not obvious whether minimizing the QAP will lead to significant improvements…”**
>
> **A6**: Solving QAPs is an NP-hard problem (Appendix C). Designing an efficient practical algorithms to QAPs is a challenging open problem on its own, which is beyond the scope of this paper. While we are currently unable to solve the QAP corresponding to Eq. (9), it is possible that solving the QAP will allow LMC to happen under weaker conditions (e.g. under a smaller network width compared to Git Re-Basin) because it enlarges the solution space compared with activation and weight matching.
>
> **Q7: “Weak additivity seems to rely heavily on the ReLU function. Does LLFC occur for other activation functions?”**
>
> **A7**: We did some preliminary experiments which showed that LMC and LLFC are difficult to achieve for other activation functions like tanh and sigmoid, so activation does play a role in LMC/LLFC. Since ReLU is the activation used in standard architectures including ResNet, VGG, etc. and is used in all previous work on LMC, an analysis based on ReLU is already highly informative given its wide adoption. We leave the exploration beyond the ReLU activation as a future direction.
>
> **Q8: “Do networks exist where LMC holds but not LLFC?…”**
>
> **A8**: No, we did not find any networks where LMC holds but not LLFC. We have conducted extensive experiments across diverse datasets, and model architectures (Sec 4 & Appendix D.2).

---

> > ### Comment · Reviewer_4GwW · 2023-08-12
> >
> > Thank you for the extensive revisions to the figures to include convincing baselines. I have increased my score accordingly.
> >
> > Despite the straightforward connection between weight/activation matching and LLFC, I concede that LLFC is a useful formalization of these heuristics. However, given that permutation methods are still basically directly optimizing for LLFC, I am still skeptical of the insights gained into LMC by this work. In particular, I would appreciate a more thorough exploration of how the theoretical findings relate to practical applications in the following points.
> >
> > 1) Weak additivity: given that you already have preliminary results on different activation functions, I would have liked to see a comparison of the LMC of networks relative to the degree to which their activation functions are weakly additive. The point is not just to use widely-adopted activation functions, but to experimentally verify that weak additivity is an important condition for LLFC and/or LMC. For example, if it were possible to get LMC with non-ReLU activations despite not having LLFC, this would be extremely informative on the reverse direction (LMC implying/not-implying LLFC).
> >
> > 2) Minimizing QAP: the point is not to solve QAP, but to ask how much of an advantage QAP has only solving the LAP for activations or weights, since in the limit (infinite width or networks that are exact permutations of one another) all of these are equivalent. In particular, the positive terms of the QAP can simply be solved as a bigger LAP, so the only question remaining is how large the negative cross terms are relatively speaking. But there may be reasons for these cross terms to be small, hence guaranteeing the effectiveness of the heuristics - e.g. conceptually weak additivity and commutativity seem to be findings in a similar vein. Alternatively, maybe the cross terms are significant and reveal where the heuristics fail. I would greatly appreciate more discussion of these cross terms in the paper.

---

> > > ### Author Response · Authors · 2023-08-21
> > >
> > > Thank you for appreciating our contributions and for raising the score!
> > >
> > >
> > > **Weak Additivity with other activations**
> > >
> > > Thank you for the suggestion. We did some preliminary experiments on different activation functions. Specifically, we conduct experiments on MLPs on MNIST with different activation functions, including ReLU, Sigmoid, and Tanh. For each model with different activation functions, we apply the spawning method to obtain two modes, $\boldsymbol{\theta_A}$ and $\boldsymbol{\theta_B}$, spawning at the same epoch. Over the test set $\mathcal{D}$, we measure the error barrier (defined in Frankle et al. [8]) between $\boldsymbol{\theta_A}$ and $\boldsymbol{\theta_B}$, and measure the LLFC and the weak additivity condition. Here, error barrier is denoted as $\text{Err}\_{Barrier}$, where $\text{Err}\_{Barrier} = \max\_{\alpha \in [0, 1]} \text{Err}\_{\mathcal{D}}(\alpha \boldsymbol{\theta}_A + (1-\alpha) \boldsymbol{\theta}_B) - \frac{1}{2}(\text{Err}\_{\mathcal{D}}(\boldsymbol{\theta}_A) + \text{Err}\_{\mathcal{D}}(\boldsymbol{\theta}_B))$. In the table below, for the cases of MLP(ReLU) and MLP(Sigmoid), the values of both $\text{Err}\_{Barrier}$ and $\mathbb{E}\_{\mathcal{D}, \ell \in [L]}[1-\text{cosine}^{(\ell)}\_{0.5}(\boldsymbol{x}_i)]$ are close to zero; for the case of MLP(Tanh), those values are not negligible. Correspondingly, the values of $\mathbb{E}\_{\mathcal{D}, \ell \in [L]}[\text{Dist}\_{\sigma}(\tilde{\boldsymbol{h}}\_{i, A}^{(\ell)},\tilde{\boldsymbol{h}}\_{i, B}^{(\ell)})]$ for MLP(ReLU) and MLP(Sigmoid) are relatively small compared to MLP(Tanh). This observation demonstrates a correlation between the onset of LMC/LLFC and the weak additivity condition, thus suggesting that weak additivity condition is important for LMC/LLFC. Also, we did not find any instance that LMC holds but LLFC doesn't.
> > >
> > > |              | $\text{Err}\_{Barrier}$ (%) | $\mathbb{E}\_{\mathcal{D}, \ell \in [L]}[1-\text{cosine}^{(\ell)}_{0.5}(\boldsymbol{x}_i)]$ | $\mathbb{E}\_{\mathcal{D}, \ell \in [L]}[\text{Dist}\_{\sigma}(\tilde{\boldsymbol{h}}\_{i, A}^{(\ell)},\tilde{\boldsymbol{h}}\_{i, B}^{(\ell)})]$ |
> > > | ------------ | -------------------------- | ------------------------------------------------------------ | ------------------------------------------------------------ |
> > > | MLP(ReLU)    | 0.115                      | 0.0385                                                       | 0.1525                                                       |
> > > | MLP(Sigmoid) | 0.060                      | 0.0203                                                       | 0.0799                                                       |
> > > | MLP(Tanh)    | **2.910**                  | **0.1157**                                                   | **0.4058**                                                   |
> > >
> > >
> > >
> > > **Minimizing QAP:…ask how much of an advantage QAP has only solving the LAP for activations or weights…**
> > >
> > > Thank you for the very interesting comments. We agree that further studying the advantages of solving the QAP over LAP is an important future direction and will make sure to discuss these points in the paper.

---

### Official Review · Reviewer_ZZ9i · 2023-07-05

**Soundness:** 3 good
**Presentation:** 3 good
**Contribution:** 3 good
**Rating:** 6
**Confidence:** 4

**Summary:**

This paper introduces Layerwise Linear Feature Connectivity (LLFC). Compared to the better known linear mode connectivity (LMC), which states that networks trained by SGD are linearly connected modulo permutation, LLFC suggests that the feature maps of every layer is connected. As shown in the paper, LLFC is a strictly stronger property than LMC. Empirical results on a range of architectures (ResNet-20, VGG-16, MLP) and datasets (MNIST, CIFAR-10) suggest that LLFC co-occurs with LMC. The authors then identify two conditions that collectively imply LLFC: weak additivity, which requires ReLU to behave like a linear activation on two modes, and commutativity, which requires the next-layer linear transformations applied to the internal features of two networks can be interchanged. They verify that these two properties holds for modes that satisfy LLFC empirically. Finally, the authors show that two common methods to obtain linearly connected modes, the spawning method and the permutation method, both promote the commutativity property, which explains their effectiveness.


**Strengths:**

- The paper reveals a novel and more general notion of linear mode connectivity, which is an interesting phenomenon that attracts recent attention in the ML community. The authors provide precise definitions and a set of sufficient conditions for LLFC. Their observation is novel and advances the understanding of the origin of linear connectivity.
- Experiments are sound and provide useful insights. In particular, empirical results support the occurrence of LLFC and validates that the two conditions for LLFC approximately hold in common settings.
- Through dissecting the conditions required for LLFC, this paper also explains why the spawning method and permutation method produces LMC. Since theoretical results are sparse in explaining the origin of LMC, this paper’s contribution on the topic is significant.
- The writing is clear and well organized.


**Weaknesses:**

- The analysis for the cause of LLFC is limited to ReLU activation. Extending the weak additivity condition to different activations does not seem straightforward. To demonstrate the prevalence of LLFC, it might help to include experiments on neural networks with different activations.
- As pointed out by the authors and shown in figure 4 and 6, weak additivity for ReLU activations and commutativity only approximately hold in real neural networks. Hence Theorem 1 describes conditions for perfect LLFC in an idealistic setting. While this result is significant and novel, a more careful treatment of the approximated version could make the theoretical contribution stronger.
- There are some gaps between theory and empirical observation. Specifically, the following results are not well explained: (a) Why are the pre-activations sparse? Is this universal to all architectures? (b) Why do the modes obtained by the spawning method share similar principal directions of model weights in each layer? While these are empirically verified in the paper, it would be helpful to point the readers to existing theoretical analysis, if available.
- While LLFC is an interesting observation and provides insights on LMC, there seems to be few applications that leverages LLFC. The idea of averaging features mentioned in the conclusion is interesting, but it is not clear how to implement feature averaging and under what situations this would be beneficial.


**Questions:**

- In the definition of LLFC, there is a scaling factor $c$ that is not predicted by theorem 1. According to the authors, this inconsistency can be attributed to the accumulation of errors in the two conditions (line 263). Why does the accumulation of errors result in a scaling difference instead of another type of modification such as an additive term?
- Does spawning or permutation have any impact on the sparsity of pre-activation $\tilde{H}^{(l)}$ and the weak additivity condition?
- Is condition 1+2 the only way to guarantee LLFC? In particular, identifying other sets of sufficient conditions for LLFC may lead to new permutation methods that find linearly connected modes, other than weight matching or activation matching.
- The experiments use SGD with momentum in training. Does the choice of optimization method affect the occurrence of LLFC?


**Limitations:**

The authors adequately addressed the limitations. There are no potential negative societal impacts of this work.

---

> ### Author Rebuttal · Authors · 2023-08-09
>
> **Q1: “The analysis for the cause of LLFC is limited to ReLU activation…To demonstrate the prevalence of LLFC, it might help to include experiments on neural networks with different activations.”**
>
> **A1**: We note that ReLU is the activation used in standard architectures including ResNet, VGG, etc., and is used in all previous work on LMC. An analysis based on ReLU is already highly informative given its wide adoption. In fact, we did some preliminary experiments which showed that LMC is difficult to achieve for some other activation functions. We leave the exploration beyond the ReLU activation as a future direction.
>
> **Q2: “While this result is significant and novel…the approximated version could make the theoretical contribution stronger.”**
>
> **A2**: Thank you for appreciating the significance and novelty of Thm 1. We agree that an approximate version could make the theoretical contribution stronger. We currently adopt the exact formulation for ease of presentation, as we believe the concise formulation already conveys the main ideas and is easier to understand and build upon.
>
> We remain open to add a careful approximate version of the theorem to the paper. Please kindly refer to our global response for more discussion on the limitation of our work.
>
> **Q3: “There are some gaps between theory and empirical observation…(a) Why are the pre-activations sparse?…(b) Why do the modes obtained by the spawning method share similar principal directions of model weights in each layer?…it would be helpful to point the readers to existing theoretical analysis, if available”**
>
> **A3**: We agree that (a) and (b) are intriguing phenomena, and theoretical explanations would be valuable. However, we are not aware of any existing theoretical analysis that's directly relevant. There are some empirical studies that bear some degree of relevance, e.g., [cite 2] empirically investigate the sparsity of the activations; [cite 1] studies the evolving statistics of model weights during the early phase of training. We agree that theoretical studies of these questions are important future directions.
>
> [cite 1] Jonathan Frankle, David J. Schwab, and Ari S. Morcos. The early phase of neural network training. In International Conference on Learning Representations, 2020.
>
> [cite 2] Torsten Hoefler, Dan Alistarh, Tal Ben-Nun, Nikoli Dryden, and Alexandra Peste. Sparsity in deep learning: Pruning and growth for efficient inference and training in neural networks. J. Mach. Learn. Res., 22(1), jan 2021. ISSN 1532-4435
>
> **Q4: “…there seems to be few applications that leverages LLFC…”**
>
> **A4**: The goal of this paper is to unveil nontrivial phenomena that offer elucidating insights into the fundamental mechanisms of deep learning. Akin to how LMC served as inspiration for applications like model soup [31], LLFC could potentially inspire applications such as feature averaging. We leave this prospect as a future direction, opting not to explore it within the scope of a single paper.
>
> **Q5: “In the definition of LLFC, there is a scaling factor $c$ that is not predicted by theorem 1…Why does the accumulation of errors result in a scaling difference instead of another type of modification such as an additive term?”**
>
> **A5**: Thank you for the great question. First of all, in most cases, we empirically observe that $c$ is close to 1 (See Appendix D.2), as predicted by Theorem 1. In other cases, we find that employing a scaling factor enables a much better description of the practical behavior than an additive error term. We will include this discussion in the paper.
>
> **Q6: “Does spawning or permutation have any impact on the sparsity of pre-activation $\tilde{\boldsymbol{H}}^{(\ell)}$ and the weak additivity condition?”**
>
> **A6**: Both the spawning and permutation methods do affect the weak additivity condition. In Figure 3 (global response), we show that the values of $\text{Dist}\_{\sigma}(\tilde{\boldsymbol{h}}\_{i, A}, \tilde{\boldsymbol{h}}\_{i, B})$ (with spawning or permutation) are negligible compared to $\text{Dist}\_{\sigma}(\tilde{\boldsymbol{h}}\_{i, C}, \tilde{\boldsymbol{h}}\_{i, D})$ (independently trained networks). Please kindly refer to the global response for more details. On the other hand, pre-activation sparsity is not affected because it is only a property of a single network.
>
> **Q7: “Is condition 1+2 the only way to guarantee LLFC? In particular, identifying other sets of sufficient conditions for LLFC may lead to new permutation methods…other than weight matching or activation matching.”**
>
> **A7**: A very interesting question. Based on our theoretical and empirical results, we believe that LLFC is strongly related to Conditions 1+2. On the other hand, while Conditions 1+2 are very insightful for our understanding of LLFC and LMC, we agree that if there exist other sufficient conditions, they might be helpful for designing new permutation methods.
>
> **Q8: “The experiments use SGD with momentum in training. Does the choice of optimization method affect the occurrence of LLFC?”**
>
> **A8**: Thank you for your question. Indeed, we have carried out experiments employing the Adam optimizer for MLPs trained on the MNIST dataset. Consequently, we believe that the choice of optimization methods does not affect the occurrence of LLFC. To further verify, we conduct additional experiments entailing the training of ResNet-20 on the CIFAR-10 dataset with Adam optimizer, as shown in Figure 1 (global response). We kindly direct you to the global response section for more experimental details.

---

> > ### Comment · Reviewer_ZZ9i · 2023-08-15
> >
> > Thank you for the detailed reply. I have read the rebuttal and other reviews and will maintain my score.

---

### Official Review · Reviewer_Dt29 · 2023-07-07

**Soundness:** 4 excellent
**Presentation:** 4 excellent
**Contribution:** 4 excellent
**Rating:** 8
**Confidence:** 3

**Summary:**

In this work, the property of Layerwise Linear Feature Connectivity (LLFC) of neural network representations is introduced, which is a stronger generalization of linear mode connectivity (LMC). They show that LLFC often occurs when LMC does. Moreover, they give a possible mechanism by which LLFC may occur (ReLU activation additivity and weight commutatitivty), and provide evidence that LLFC often does occur by this mechanism. Finally, they reinterpret spawning and permutation-finding based methods for LMC as promoting commutativity.


**Strengths:**

1. Introduction exposition is quite good.
2. The relationship between spawning / permutation-finding methods and commutatitivity is really interesting and insightful. The fact that both the weight-matching and activation-matching losses can be found in the commutativity condition is quite neat, as (to the best of my knowledge) these two methods were proposed as heuristics. The discussion of weight matrix rank in Appendix E is also insightful.
3. Figure 5 on the empirical measurement of commutativity is very nice, especially because the two baseline numbers (weight distance and activation distance) are so large, so commutatitivity really seems to be a good thing to look at here.
4. Although LLFC is stronger than LMC, in some sense it may make the study of LMC easier, because you have more to look at (features in each layer and the two sufficient conditions you give, not just loss of a whole neural net).



**Weaknesses:**

1. You don't really cover the last method (straight-through estimator) from Git Re-Basin, even though it works the best in many cases. Perhaps you should note this.
2. Although these experimental setups are standard, it would be good to see to what extent this holds beyond these few architectures for image classification.


**Questions:**

Typos:
1. In 5.2, (i) should have $W_{B, \mathrm{pri}}^{(l)} H^{(l-1)}_A + W_{A, \mathrm{pri}}^{(l)} H_B^{(l-1)}$.
2. 5.3: "conprehensive" -> "comprehensive"


**Limitations:**

I do not see much discussion of limitations, besides on the hardness of exactly minimizing the commutativity property objective. Perhaps you could state that the empirical evidence is limited to image classification.

---

> ### Author Rebuttal · Authors · 2023-08-09
>
> **Q1: Ask for additional experiments to cover the straight-through estimator method from Git Re-basin [1]. “You don't really cover the last method (straight-through estimator) from Git Re-Basin, even though it works the best in many cases. Perhaps you should note this.”**
>
> **A1**: We greatly appreciate your suggestions. We follow your suggestions and conduct new experiments to cover the Straight-Through Estimator (STE) method.
>
> Our experiments follow the same settings as in Ainsworth et al. [1], which applied STE method on MLPs trained on MNIST and CIFAR-10 datasets. Correspondingly, in congruence with the experiments in Section 4 of the main paper, we measure both ${\rm cosine}\_{\alpha}(\boldsymbol{x}_i)$ and ${\rm cosine}\_{A, B}(\boldsymbol{x}_i)$ for each data point $\boldsymbol{x}_i$ in the test set $\mathcal{D}$. In Figure 1 (global response), the values of $\mathbb{E}\_{\mathcal{D}}[1-{\rm cosine}\_{\alpha}(\boldsymbol{x}_i)]$ are close to zero compared to $\mathbb{E}\_{\mathcal{D}}[1-{\rm cosine}\_{A,B}(\boldsymbol{x}_i)]$ across different layers, datasets, and different values of $\alpha$, which further convincingly verifies that LLFC consistently co-occurs with LMC. Please kindly refer to the global response for more experimental details.
>
> **Q2: Suggestions for empirical evidence beyond image classification. “Although these experimental setups are standard, it would be good to see to what extent this holds beyond these few architectures for image classification.”**
>
> **A2**: Thank you for your suggestions. We acknowledge that our experiments follow the standard setups in the LMC literature. We will leave the exploration beyond image classification as future directions. We have included this in the list of limitations (see global response).
>
> **Q3: Typos.**
>
> **A3**: Thank you for spotting the typos! We will fix them.
>
> **Q4: Ask for more discussion about the limitation of this paper. “I do not see much discussion of limitations, besides on the hardness of exactly minimizing the commutativity property objective. Perhaps you could state that the empirical evidence is limited to image classification.”**
>
> **A4**: Thank you for the suggestion. Please refer to the global response for a list of limitations, which we will add to the paper.

---

> > ### Comment · Reviewer_Dt29 · 2023-08-16
> >
> > Hello,
> >
> > Thank you for your reply. It is great that you include STE experiments and limitations now! I would like to retain my score.

---

### Official Review · Reviewer_6KFq · 2023-07-07

**Soundness:** 3 good
**Presentation:** 3 good
**Contribution:** 3 good
**Rating:** 7
**Confidence:** 3

**Summary:**

This paper presents a special case of Linear Mode Connectivity (LMC) denoted as Layerwise Linear Feature Connectivity (LLFC). Whereas two trained neural networks present LMC if a convex combination of their parameters produce a neural network with similar training loss and accuracy, two trained neural networks present LLFC if a convex combination of their features in every layer also produces a neural network with similar features. Although being a special case, the authors find out that LMC and LLFC co-occur very often. Moreover, the authors characterize conditions that jointly imply LLFC between two ReLU networks.

*****

Following the authors' response, I am updating my score from 6 to 7.

**Strengths:**

The writing of the paper is very clear and the experiments are distributed in such a way along the text that it is easy to follow along.

Moreover, I find the study of how this conditions may emerge in ReLU networks particularly relevant, since in that simpler setting it is easier to understand what is going on.

**Weaknesses:**

The authors sell the idea of LLFC in a very positive tone, which is actually quite common in papers, but I cannot help but wonder about the following:

Two neural networks presenting LLFC would have very similar parameters, and the fact that LLFC co-occurs with LMC implies that LMC is observed due to the similarity between trained neural networks - either because the first epochs of training determined what model would be ultimately obtained; or because there are relatively few optimal neural networks upon permutation.

However, even under this interpretation, I believe that this study helps demystifying LMC if it turns out that LMC rarely occur without LLFC.

**Questions:**

The introduction attributes the first observation of LMC to [8], but Section 2 attributes it to earlier work [21].

Definition 2 needs rework because it is clearly invalid if $\alpha=0$ or $\alpha=1$ unless $c=1$. In fact, this is not even how you measure it in Section 4, since you use cosine similarity instead. I believe that the use of a tolerance factor  $\epsilon$ like you did in Line 220 would be more adequate in this definition.

Would it make sense to assume that Definition 3 has some connection with stable neurons in ReLU networks, as in the following papers: https://arxiv.org/abs/1711.07356 & https://arxiv.org/abs/2102.07804 ?

It seems to me that Condition 1 was inspired by spawning and Condition 2 by permutation, even though they are not worded in that way. Is that a valid way to interpret them?

I would like to hear the thoughts of the authors about what I described in Weaknesses.

**Limitations:**

I could not identify a discussion about limitations in the paper.

---

> ### Author Rebuttal · Authors · 2023-08-09
>
> **Q1: Concerns about the trivial case that two Neural Networks (NNs) emerging LLFC share the similar weights. “ I cannot help but wonder about the following: Two neural networks presenting LLFC would have very similar parameters, and the fact that LLFC co-occurs with LMC implies that LMC is observed due to the similarity between trained neural networks”**
>
> **A1**: Thank you for your great question. In fact, our experiments have already ruled out the trivial case that two NNs share similar weights.
>
> First, we directly evaluated the difference between the weights of two NNs that yield LLFC, namely $\text{Dist}_W = \text{dist}\left(\text{vec}({\boldsymbol{W}}_A^{(\ell)}), \text{vec}({\boldsymbol{W}}_B^{(\ell)})\right)$, where $\text{dist}(\boldsymbol{x}, \boldsymbol{y}) := \|\boldsymbol{x} - \boldsymbol{y}\|^2 / (\| \boldsymbol{x}\| \cdot \|\boldsymbol{y}\|)$. In Figure 5 (main paper), the values of $\text{Dist}\_W$ are usually within 0.5~1.5 while the values of $\text{Dist}\_{com}$ are close to zero.
>
> Second, we calculated the cosine similarity between the features of two linearly connected NNs, namely, ${\rm cosine}\_{A, B}(\boldsymbol{x}_i)$. If the weights of two NNs are similar, their features should display similarity as well. Nevertheless, in Figure 2 and 3 (main paper), the value of $\mathbb{E}\_{\mathcal{D}}[1-{\rm cosine}\_{A, B}(\boldsymbol{x}_i)]$ could reach its maximum at around 0.75.
>
> Consequently, we can confidently dismiss the trivial case that two NNs emerging LLFC share similar weights.
>
> **Q2: Whether LMC rarely occur without LLFC. “However, even under this interpretation, I believe that this study helps demystifying LMC if it turns out that LMC rarely occur without LLFC.”**
>
> **A2**: Thank you for acknowledging our contribution. We would like to clarify that we did not find any instances where two NNs exhibit LMC but not LLFC. We conducted extensive experiments across diverse datasets, network architectures, and various layers within the network (Sec 4 & Appendix D.2). Therefore, we believe LLFC is a more fundamental phenomenon that helps demystify LMC.
>
> **Q3: A reference problem. “The introduction attributes the first observation of LMC to [8], but Section 2 attributes it to earlier work [21].”**
>
> **A3**: Thank you for pointing out this problem.  In fact, [8] references [21] and attributes the initial observation to [21]. However, [8] was the first to formally define and thoroughly investigate the LMC problem. Thank you for your careful review, we will later clarify this point in the revised version of our paper.
>
> **Q4: Concerns about Definition 2. “Definition 2 needs rework because it is clearly invalid if $\alpha =0$ or $\alpha =1$ unless $c=1$.”**
>
> **A4**: Thank you for your careful review again. In Definition 2, $c$ is allowed to depend on the interpolation parameter $\alpha$, i.e., "$\forall \alpha \in [0, 1], \exists c > 0$". When $\alpha = 0$ or $\alpha = 1$, $c=1$ will satisfy the condition. Consequently, the definition remains valid even at the boundary cases of $\alpha = 0$ or $\alpha = 1$. For other values of $\alpha$, $c$ can be different from $1$. Thus, using cosine similarity to verify LLFC (Definition 2) is appropriate.
>
> **Q5: Where Definition 3 has some connection with stable neurons in ReLU networks. “Would it make sense to assume that Definition 3 has some connection with stable neurons in ReLU networks…?”**
>
> **A5**: Thank you for your question. Regarding the two papers you referenced, stable neuron is defined as one whose output is the constant value zero ($y=0$) or the pre-activation output ($y=x$) on all inputs, which is a property concerning a single network. On the other hand, Definition 3 concerns a relation between two networks. Therefore, these two properties are orthogonal to each other, and their connection is not clear. That said, both properties describe interesting linearity phenomena in ReLU networks, and we will make a note of this in the paper.
>
> **Q6: Question about the inspiration and interpretation of Condition 1 and 2. “It seems to me that Condition 1 was inspired by spawning and Condition 2 by permutation, even though they are not worded in that way. Is that a valid way to interpret them?”**
>
> **A6**:  We are happy to explain what inspired Conditions 1&2. In fact, they were not inspired from spawning or permutation methods but stemmed from the derivation process of Theorem 1. Through this, we identified that the two conditions facilitate the derivation of the feature connectivity across layers. Subsequently, we discovered the connection between Condition 2 and both spawning and permutation methods. This intriguing connection leads us to the conjecture that both spawning method and permutation method essentially contribute to the fulfillment of LLFC.
>
> **Q8: Ask for more discussion about the limitations of this paper. “I could not identify a discussion about limitations in the paper.”**
>
> **A8**: We are happy to discuss the limitations of this paper. Please kindly refer to the global response for a list of limitations, which we will add to the paper.

---

> > ### Comment · Reviewer_6KFq · 2023-08-13
> >
> > I appreciate and am satisfied with the responses from the authors. I am updating my score as a reflection of that.

---

### Author Rebuttal · Authors · 2023-08-09

**Limitations.**

1. In Appendix C, identifying a permutation that directly enforces commutativity condition involves solving a NP-hard QAP. We leave the QAP-solving problem as a future direction.
2. Our Theorem 1 predicts LLFC in an ideal case, while in practice, a scaling factor $c$ is introduced to the definition of LLFC to better describe the experimental results. Realistic theorems and definitions (approximated version) are deferred to future research.
3. Our current experiments mainly focus on image classification, aligning with existing literature on LMC. While we appreciate suggestions to extend empirical evidence beyond image classification, we commit to exploring this avenue in future research.

**Experimental details.**

**[Figure 1] Further verify LLFC under various settings.**

To verify the LLFC property, we measure ${\rm cosine}\_{\alpha}(\boldsymbol{x}_i)$ and compare with ${\rm cosine}\_{A, B}(\boldsymbol{x}_i)$, consistently with main paper. In Fig 1, the values of $\mathbb{E}\_{\mathcal{D}}[1-{\rm cosine}\_{\alpha}(\boldsymbol{x}_i)]$ are close to 0 compared with $\mathbb{E}\_{\mathcal{D}}[1-\text{cosine}\_{A, B}(\boldsymbol{x}_i)]$, and thus convincingly verify our claim.

Notably, all the experimental settings are standard: the settings of MLPs on the MNIST and CIFAR-10 follows Ainsworth et al. 2022 [1]; the training of ResNet-20 on the CIFAR-10 follows the default settings of Pytorch; the training of ResNet-50 on the Tiny ImageNet follows Frankle et al. [8].

**[Figure 2] Add baseline of non-LMC models for verifying LLFC.**

We measure ${\rm cosine}_{0.5}(\boldsymbol{x}_i)$ on both the linearly connected models and independently trained models, denoted as ${\rm cosine}\_{LMC}(\boldsymbol{x}_i)$ and ${\rm cosine}\_{not \ LMC}(\boldsymbol{x}_i)$ correspondingly. In Fig 2, the values of $\mathbb{E}\_{\mathcal{D}}[1-{\rm cosine}\_{LMC}(\boldsymbol{x}_i)]$ are negligible compared to $\mathbb{E}\_{\mathcal{D}}[1-{\rm cosine}\_{not \ LMC}(\boldsymbol{x}_i)]$. Therefore, we rule out the possibility that LLFC is a byproduct of averaging.

**[Figure 3] Add baselines for verifying weak additivity and sparsity.**

First, to verify the weak additivity , we compared ${\rm Dist}\_{\sigma}(\tilde{\boldsymbol{h}}\_{i, A}, \tilde{\boldsymbol{h}}\_{i, B})$ with ${\rm Dist}\_{\sigma}(\tilde{\boldsymbol{h}}\_{i, C}, \tilde{\boldsymbol{h}}\_{i, D})$ and ${\rm Dist}\_{\sigma}(\boldsymbol{r}_1, \boldsymbol{r}_2)$. The subscripts $A, B, C, D$ denote four different models, where $A, B$ are linearly connected and $C, D$ are independently trained. In Fig 3, the values of ${\rm Dist}\_{\sigma}(\tilde{\boldsymbol{h}}\_{i, A}, \tilde{\boldsymbol{h}}\_{i, B})$ are negligible compared with ${\rm Dist}\_{\sigma}(\tilde{\boldsymbol{h}}\_{i, C}, \tilde{\boldsymbol{h}}\_{i, D})$ and ${\rm Dist}\_{\sigma}(\boldsymbol{r}_1, \boldsymbol{r}_2)$. Therefore, we verify that the weak additivity condition holds for modes that are linearly connected.

Second, to verify the sparsity claim, we measure the sparsity of the pre-activations of both well-trained networks and random initialized networks, using $S(\boldsymbol{x}) = \frac{\|\boldsymbol{x}\|_1}{n\|\boldsymbol{x}\|\_{\infty}}(\boldsymbol{x} \in \mathbb{R}^n)$, denoted as $S(\tilde{\boldsymbol{h}}\_{i, \text{end}})$ and $S(\tilde{\boldsymbol{h}}\_{i, \text{init}})$ respectively. In Fig 3, the majority of the values of $S(\tilde{\boldsymbol{h}}\_{i, \text{end}})$ are distinctively smaller than $S(\tilde{\boldsymbol{h}}\_{i, \text{init}})$. Therefore, we validate our sparsity claim.

Notably, in Fig 3, the values of ${\rm Dist}\_{\sigma}(\tilde{\boldsymbol{h}}\_{i, C}, \tilde{\boldsymbol{h}}\_{i, D})$ are smaller than ${\rm Dist}\_{\sigma}(\boldsymbol{r}_1, \boldsymbol{r}_2)$. The gap between between ${\rm Dist}\_{\sigma}(\tilde{\boldsymbol{h}}\_{i, C}, \tilde{\boldsymbol{h}}\_{i, D})$ and ${\rm Dist}\_{\sigma}(\boldsymbol{r}_1, \boldsymbol{r}_2)$ increases as the gap between the gap between $S(\tilde{\boldsymbol{h}}\_{i, \text{end}})$ and $S(\tilde{\boldsymbol{h}}\_{i, \text{init}})$ widens. This observation supports that the weak additivity is likely to hold if the pre-activations are sparse enough.

**[Figure 4] Add baselines of non-LMC models for verifying commutativity.**

We measure ${\rm Dist}\_{com}$ on both the linearly connected models and independently trained models, denoted as ${\rm Dist}\_{com, LMC}$ and ${\rm Dist}\_{com, not\ LMC}$. In Fig 4, the values of ${\rm Dist}\_{com, LMC}$ are negligible compared with ${\rm Dist}\_{com, not\ LMC}$, which validates the commutativity condition for LMC models.

**[Figure 5] Verify the top singular vectors of $V$ hold a small principal angle similar to $U$.**

We extract the top $k$ singular vectors from $V_A$ and $V_B$ and compute the minimal principal angle $\beta_{V}$ between the subspaces spanned by these singular vectors. We compare the principal angle of two linearly connected models, $\beta_{LMC, V}$, with that of two independently trained models, $\beta_{not \ LMC, V}$. In Fig 5, the angle $1-\cos\beta_{LMC, V}$ is close to 0 while $1-\cos\beta_{not\ LMC, V}$ is significantly large, thus confirming our claim.

**[Figure 6] Relationship between shared early training epochs and the similarity between principal directions of spawned models.**

We compute both $1-\cos\beta_{U}$ and $1-\cos\beta_{V}$ for two models that spawning at different iteration $t$.  In Fig 6, across different layers, both $1-\cos\beta_{U}$ and $1-\cos\beta_{V}$ decrease as the spawning iteration increases. This implies that similarity between principal directions of spawned models increases if sharing more early training iterations.

Furthermore, a noticeable similarity emerges between the curves of $1-\cos\beta$ v.s. spawning iteration $t$ and the curves of the instability (defined in Frankle et al. [8]) v.s. the spawning iteration $t$ (Fig 3 in Frankle et al. [8]). This enhances the credibility of our analysis significantly.

---

> ### Comment · Area_Chair_zxXP · 2023-08-18
> **Thank you for the rebuttal**
>
> Dear authors,
>
> thank you for providing a rebuttal, I will take it into account when giving my recommendation.
>
> Best,
> Your AC

---

### Decision · Program_Chairs · 2023-09-21

**Decision:**

Accept (poster)

**Comment:**

This paper considers the phenomenon of linear model connectivity in trained neural networks. The main contribution is to provide empirical evidence of an even stronger notion of linear connectivity: Layerwise Linear Feature Connectivity (LLFC). This means that the feature maps of every layer can be connected.

The reviewers agree that this is an interesting contribution and the very detailed response of the authors has helped resolve most of the issues raised in the initial reviews. After the rebuttal, there is a clear consensus towards accepting the paper. After my own reading of the reviews, rebuttal and paper, I agree with this view and recommend acceptance. I warmly encourage the authors to incorporate the additional numerical results, experimental details and discussions from the rebuttal phase in the final version.